# Evaluating the potential of Vitamin D and curcumin to alleviate inflammation and mitigate the progression of osteoarthritis through their effects on human chondrocytes: A proof-of-concept investigation

Rajashree Patnaik[1], Sumbal Riaz[1], Bala Mohan Sivani[1¤], Shemima Faisal[1], Nerissa Naidoo[1], Manfredi Rizzo[2]*, Yajnavalka Banerjee[1,3]*

1 College of Medicine and Health Sciences, Mohammed Bin Rashid University of Medicine, and Health Sciences (MBRU), Dubai, United Arab Emirates, 2 Department of Health Promotion, Mother and Child Care, Internal Medicine, and Medical Specialties (Promise), University of Palermo, Palermo, Italy, 3 Centre for Medical Education, University of Dundee, Dundee, United Kingdom

¤ Current address: Department of Molecular Biology, Lund University, Lund, Sweden
* yajnavalka.banerjee@mbru.ac.ae (YB); manfredi.rizzo@unipa.it (MR)

**Data Availability Statement:** All our data are within the paper, and uploaded the same on https://

## Abstract

Osteoarthritis (OA) is a chronic degenerative joint disorder primarily affecting the elderly, characterized by a prominent inflammatory component. The long-term side effects associated with current therapeutic approaches necessitate the development of safer and more efficacious alternatives. Nutraceuticals, such as Vitamin D and curcumin, present promising therapeutic potentials due to their safety, efficacy, and cost-effectiveness. In this study, we utilized a proinflammatory human chondrocyte model of OA to assess the anti-inflammatory properties of Vitamin D and curcumin, with a particular focus on the Protease-Activated Receptor-2 (PAR-2) mediated inflammatory pathway. Employing a robust siRNA approach, we effectively modulated the expression of PAR-2 to understand its role in the inflammatory process. Our results reveal that both Vitamin D and curcumin attenuate the expression of PAR-2, leading to a reduction in the downstream proinflammatory cytokines, such as Tumor Necrosis Factor-alpha (TNF-α), Interleukin 6 (IL-6), and Interleukin 8 (IL-8), implicated in the OA pathogenesis. Concurrently, these compounds suppressed the expression of Receptor Activator of Nuclear Factor kappa-B Ligand (RANKL) and its receptor RANK, which are associated with PAR-2 mediated TNF-α stimulation. Additionally, Vitamin D and curcumin downregulated the expression of Interferon gamma (IFN-γ), known to elevate RANKL levels, underscoring their potential therapeutic implications in OA. This study, for the first time, provides evidence of the mitigating effect of Vitamin D and curcumin on PAR-2 mediated inflammation, employing an siRNA approach in OA. Thus, our findings pave the way for future research and the development of novel, safer, and more effective therapeutic strategies for managing OA.

figshare.com/ and have been provided with the Digital object identifier (DOI) 10.6084/m9.figshare. 23623059.

**Funding:** This work was made possible by the generous support from Mohammed Bin Rashid University of Medicine and Health Sciences (MBRU), with a research award, identifier MBRU-CM-RG2022-04, granted to both NN and YB. Further support was provided to SR in the form of a scholarship within the Biomedical Sciences Master Program at MBRU. It is important to note that the funders did not have any involvement in the study's design, data collection and analysis, the decision to publish, or the preparation of the manuscript.

## Introduction

Osteoarthritis (OA) is a complex disease impacting various joint structures, originally attributed to "wear and tear" [1]. Affecting 10% of men and 18% of women above 60 [2], its primary risk factor is aging, with rapid prevalence increase post-50 [3]. Genetics, systemic factors, biomechanics, and past injury also contribute to OA onset [2, 4–6]. Knee OA is notably prevalent in the US, impacting up to 14 million people, half of whom are under 65 [7]. OA also disproportionately affects women with concurrent hand and knee OA [8].

OA's economic burden is immense, resulting in wage losses of $65 billion and medical expenses surpassing $100 billion [9]. Comorbidities are common in OA patients, with 31% having 5 or more additional conditions [10]. OA patients exhibit a 20% increased mortality rate due to reduced physical activity [11].

OA pathophysiology implicates the entire joint, with cartilage degradation resulting from biomechanical stress and possible genetic predisposition causing early chondrocyte damage [12]. Prolonged OA progression leads to irreversible chondrocyte depletion and matrix degradation, with structural changes involving the joint's [12].

OA treatment primarily encompasses non-pharmacological approaches, including physical therapies and lifestyle modifications [13]. Pharmacological measures include topical analgesics, non-steroidal anti-inflammatory drugs (NSAIDs), intra-articular hyaluronans, and more [14].

Recent advancements focus on counterbalancing the cartilage-deteriorating enzymatic activity that characterizes OA [15]. Examples include Sprifermin, a synthetic form of human fibroblast growth factor (FGF)-18, that promotes cartilage growth [16, 17] and drugs targeting matrix metalloproteinases (MMPs) that degrade cartilage [18, 19]. Similarly, a disintegrin and metalloproteinase (ADAM) metallopeptidase with Thrombospondin Type 1 Motif 5 (ADAMTS-5) inhibitors and Wingless/integrated (Wnt) inhibitors show promise [20–25].

Inflammatory pathways play a crucial role in OA development, with chondrocytes producing pro-inflammatory cytokines under stress, contributing to cartilage degradation and eventual OA onset [26–28]. Several clinical trials have investigated the potential of drugs targeting proinflammatory cytokines [29–31]. Data from these trials suggest that drugs targeting proinflammatory cytokines show promise in modulating immune response and reducing inflammation-related symptoms, potentially leading to improved patient outcomes. However, the therapeutic efficiency varies, and further studies are required to confirm their long-term safety and efficacy.

Given the significant burden and limitations of current OA treatments, further research into effective, side-effect-minimal therapies is necessary. Nutraceuticals, such as Vitamin D and curcumin, present potential therapeutic alternatives, offering anti-inflammatory effects, potentially addressing OA inflammation [32, 33]. The variable results of Vitamin D in OA management (reviewed and summarized in Table 1), and the promising but preliminary data on curcumin highlight the need for further investigation [34, 35].

In this innovative *in vitro* study, we delve into the potential anti-inflammatory effects of Vitamin D (specifically, 1,25-dihydroxyVitamin D3) and curcumin (1,6-heptadiene-3,5-dione-1,7-bis(4-hydroxy-3-methoxyphenyl)-(1E,6E)) within a proinflammatory OA chondrocyte model. Our focus is on the reduction of protease activated receptor– 2 (PAR– 2) triggered inflammation, a significant feature of OA. The results showcase considerable anti-inflammatory responses from both nutraceuticals, offering innovative insights into their possible role in curtailing PAR-2 induced inflammation in OA. This study presents robust evidence that Vitamin D and curcumin might represent a pioneering, natural, and efficacious therapeutic strategy for managing OA and mitigating its related symptoms, specifically those exacerbated by PAR-2 signaling.

**Table 1. Studies assessing the effect of Vitamin D in OA.**

| S/N | Number of subjects | Inclusion criteria | Completion year | Primary outcome | Secondary outcome | Trial/Study Type | Follow-up | Observed Effect | Reference |
|---|---|---|---|---|---|---|---|---|---|
| 1 | 146 | Chronic knee discomfort, WOMAC score of ≥ 1, K & L score of ≥ 2 on knee radiograph, knee pain on examination | 2009 | Cartilage volume loss-by MRI knee pain severity- WOMAC pain scale, 0–20:0, no pain; 20, extreme pain. | Physical function, knee function (WOMAC function scale, 0–68: 0, no difficulty; 68,extreme difficulty), cartilage thickness, bone marrow lesions, and radiographic joint space width. | 2 | 2 years | No beneficial effect was observed | [36] |
| 2 | 474 | Age >50 years, radiological evidence of knee OA, K&L score 2/3, JSW >1mm, and knee pain for most days of the previous month. | 2016 | Primary outcome was rate of JSN in medial compartment of knee. | Rates of JSN in lateral compartments, K&L grade, WOMAC pain, stiffness, function and "get up and go test". | N/A | 3 years | No beneficial effect was observed | [37] |
| 3 | 1,398 | Men aged ≥50 and women aged ≥55 with chronic knee pain including those with prior unilateral total knee replacement. | 2018 | Serum Levels of Biomarkers of Systemic Inflammation, IL6, CRP, TNFR2. Incident Autoimmune Diseases. Severity of Knee Pain in Subsample with chronic, frequent knee pain at baseline- with n-3 fatty acid & Vitamin D. | Incident Autoimmune Disease | 3 | 6 years | No beneficial effect was observed | [38] |
| 4 | 340 | Participants with symptomatic knee osteoarthritis for at least 6 months (using ACR criteria), pain of >20mm on a 100-mm VAS scale and Vitamin D insufficiency (between 12.5 and 60 nmol/L) | 2016 | Knee cartilage volume, cartilage defects, bone marrow lesions, and effusion synovitis volume using MRI | Knee symptoms using WOMAC arthritis index | N/A | 2 years | Positive outcome was observed | [39] |
| 5 | 200 | Participants with symptomatic knee osteoarthritis for at least 6 months (using ACR criteria), pain of >20mm on a 100-mm VAS scale and Vitamin D insufficiency (between 12.5 and 60 nmol/L) | 2018 | Change in serum high-sensitive C-reactive protein, IL-6, IL-8, IL-10, leptin, adiponectin, resistin, adipsin and apelin were measured. | Knee pain assessed by WOMAC arthritis index, tibial cartilage volume using MRI, joint effusion synovitis using MRI | N/A | 2 years | No beneficial effect was observed | [40] |
| 6 | 107 | Participants ≥50 with crepitus on knee ROM, morning stiffness of short duration, knee pain for 6 months and WOMAC pain score greater than 4 on the Likert scale been receiving conventional treatment for OA for at least 6 months; no BMI greater than 30 kg/m²; no previous fractures of the index knee; no previous surgeries on the index knee; no secondary OA and no allergies to any of the substances used | 2013 | Knee pain (WOMAC and VAS) and knee function (WOMAC) | changes in biochemical parameters (serum calcium [total and ionic], serum phosphorus, 25 (OH)D, and alkaline phosphatase) | NA | 1 year | Positive outcome was observed | [41] |

(*Continued*)

**Table 1.** (Continued)

| S/N | Number of subjects | Inclusion criteria | Completion year | Primary outcome | Secondary outcome | Trial/ Study Type | Follow-up | Observed Effect | Reference |
|---|---|---|---|---|---|---|---|---|---|
| 7 | 340 | Subjects aged 50 to 79 years, with symptomatic knee osteoarthritis (OA) with a pain visual analogue scale (VAS) of at least 20 mm in most days of the last month; Have an American College of Rheumatology (ACR) functional class rating of I, II and III. Have relatively good health. Have serum Vitamin D level of >12.5 nmol/L and <60 nmol/L | 2016 | Knee effusion-synovitis volume in suprapatellar and other regions by MRI | | N/A | 2 years | Positive outcome was observed | [42] |
| 8 | 340 | Patients Aged 50–79 years, with symptomatic knee osteoarthritis (OA) with a pain visual analogue scale (VAS) of at least 20 mm in most days of the last month; Have an American College of Rheumatology (ACR) functional class rating of I, II and III. Have relatively good health. Have serum Vitamin D level of >12.5 nmol/L and <60 nmol/L | 2013 | Change in tibial cartilage volume (assessed using MRI) and change in the WOMAC pain score (0 = no pain, 500 = most pain) | Cartilage defects and bone marrow lesions (assessed using MRI). | 3 | 2 years | No beneficial effect was observed | [43] |
| 9 | 50 | K&L score of 1–4 in the medial tibio-femoral knee compartment, JSW> 1 mm and knee pain for most days of the previous month | 2019 | Rate of joint space narrowing assessed using plain radiographs (STV, subchondral BML volume) and knee symptoms assessed using the WOMAC questionnaire with each item scored on a VAS | Blood samples to measure serum Vitamin D3 at baseline, 12 and 36 months, serial MRIs at baseline and at 12 monthly intervals | NA | 2 years | No beneficial effect was observed | [44] |
| 10 | 287 | Seniors aged ≥60 years undergoing elective surgery for unilateral total knee replacement due to severe knee osteoarthritis | 2014 | Pain and disability related to rehabilitation of the operated knee assessed using WOMAC and related to the expected high prevalence of OA in the contra-lateral knee | change in 25(OH)D levels, muscle strength, balance, lower extremity function, the rate of falls, bone density and bone quality, bone metabolism, general pain, fall-related fractures | 4 | 2 years | No beneficial effect was observed | [45] |
| 11 | 1189 | RCTs that compared Vitamin D (in any form and dose) with placebo in patients with knee OA aged 45 years and above | 2016 | Change in Western Ontario and McMaster Universities (WOMAC) index, Visual Analog Scale (VAS) and Functional Pain Score (FPS) | Change in tibial cartilage thickness, joint space width and safety profile | SR | N/A | No beneficial effect was observed | [46] |
| 12 | 413 | All randomized, cohort, or case-controlled studies reported in English of adults undergoing THR/TKR where Vitamin D supplementation was given peri-operatively and at least one outcome was reported were included | 2021 | WOMAC scores for both operated and non-operated knees, and the rate of falls over 24 months | A sit-to-stand test, 4-m normal gait speed, activity level, and radiographic progression in the contralateral knee | SR | N/A | No beneficial effect was observed | [47] |

(*Continued*)

**Table 1.** (Continued)

| S/N | Number of subjects | Inclusion criteria | Completion year | Primary outcome | Secondary outcome | Trial/ Study Type | Follow-up | Observed Effect | Reference |
|---|---|---|---|---|---|---|---|---|---|
| 13 | N/A | (i) any cohort, case control or cross-sectional studies examining the association of 25-(OH)D levels with OA outcomes (including symptomatic, radiographic or structural changes) in adults and (ii) randomized controlled trials (RCTs) examining Vitamin D supplementation (all forms and all doses) for OA. | 2013 | Knee radiographic OA as assessed by the Kellgren and Lawrence (KL) score | Joint space narrowing and changes in cartilage volume | SR | N/A | Positive outcome was observed | [48] |
| 14 | N/A | (1) study design: RCTs; (2) population: patients diagnosed with keen OA; (3) intervention: Vitamin D supplementation; (4) comparison: placebo; (5) outcome measure: changes in WOMAC pain, WOMAC stiffness, WOMAC function, changes from baseline in Vitamin D3 level, changes in tibial cartilage volume (mm3), and adverse events. | 2017 | N/A | N/A | MA | N/A | Positive outcome was observed | [49] |
| 15 | N/A | Cross-sectional, cohort, and case-control studies, reporting RR, OR, CI related to Vitamin D and risks of knee OA, RCTs studies investigating the difference in change rates of WOMAC pain scores, tibial cartilage volume, and JSW between knee OA patients given Vitamin D and patients given placebo | 2019 | N/A | N/A | MA | N/A | No beneficial effect was observed | [50] |
| 16 | 2983 | Literature search of PubMed, EMBASE and Cochrane Library for studies that addressed the association between VDR polymorphisms and OA until March 2020. Studies were selected based on (i) a genotype-based study, focusing on any of the four VDR polymorphisms: ApaI, BsmI, TaqI and FokI; (ii) patients with primary or idiopathic OA diagnosed according to ACR clinical OA criteria, imaging or total joint replacement due to primary OA; (iii) the study provided sufficient data to calculate the OR and the corresponding 95% CI; (iv) genotype frequencies did not significantly deviate from HWE; and (v) there was a case–control or cohort design | 2020 | N/A | N/A | MA | N/A | Positive outcome was observed | [51] |

*(Continued)*

**Table 1.** (Continued)

| S/N | Number of subjects | Inclusion criteria | Completion year | Primary outcome | Secondary outcome | Trial/Study Type | Follow-up | Observed Effect | Reference |
|---|---|---|---|---|---|---|---|---|---|
| 17 | 1599 | Inclusion criteria: (1) study design: RCTs; (2) population: patients diagnosed with KOA with associated symptoms and older than 45; (3) intervention: Vitamin D supplements; (4) comparison: placebo; (5) outcome index: the main outcome indicators were WOMAC scores, volume of tibia cartilage, bone marrow lesions, joint space width and synovial fluid volume | 2020 | N/A | N/A | MA | N/A | Positive outcome was observed | [52] |
| 18 | 2,593 | Study/studies that–(a) measured Vitamin D status in patients undergoing hip or knee joint surgery; (b) reported prevalence of Vitamin D status; (c) reported outcomes of assessments with 1 or more relevant validated tools by comparing hypovitaminosis D and euvitaminosis D groups; and (d) reported preoperative as well as postoperative scores of assessment tools. 933males with age 69.89 were included | 2018 | N/A | N/A | MA | N/A | Positive outcome was observed | [53] |
| 19 | N/A | 25(OH) Vitamin D serum levels were measured, and prevalence, incidence, and progression of OA of knees, hips and hands was scored by the K & L grading system. | 2016 | N/A | N/A | MA | N/A | No beneficial effect was observed | [54] |
| 20 | 1626 | (a) genotype based focusing on any of the three VDR polymorphisms: BsmI, TaqI, and ApaI; (b) study investigating primary or idiopathic OA; (c) data regarding genotype distributions; (d) The VDR genotype frequencies not significantly deviating from HWE were included | 2014 | N/A | N/A | MA | | No beneficial effect was observed | [55] |

(*Continued*)

**Table 1.** (Continued)

| S/N | Number of subjects | Inclusion criteria | Completion year | Primary outcome | Secondary outcome | Trial/ Study Type | Follow-up | Observed Effect | Reference |
|---|---|---|---|---|---|---|---|---|---|
| 21 | 2104 | Patients with OA diagnosed using ACR clinical OA criteria, imaging (e.g., Schneiderman or K & L grade system or total joint replacement due to primary OA | 2014 | N/A | N/A | MA | N/A | Positive outcome was observed | [56] |

SR = Systematic Review; MA = Meta-analysis; N/A = Not Applicable

Abbreviations

ACR criteria-American College of rheumatology

BML volume-Bone marrow lesion volume

CI- confidence interval

CENTRAL- Cochrane Central Register of Controlled Trials

CNKI- China National Knowledge Infrastructure

HWE-Hardy Weinberg Equilibrium

JSN-joint space narrowing

JSW- joint space width, JSN- joint space narrowing

MFPDI- Manchester Foot Pain and Disability Index

MRI- magnetic resonance imaging

OA-osteoarthritis

OR-odds ratio

RCT-randomized controlled trials

ROA-radiographic osteoarthritis

RR-relative risk

STV- synovial tissue volume

TKR- total knee replacement, THR-total hip replacement

VAS-visual analogue scale

WOMAC pain scale-Western Ontario and McMaster Universities pain scale, K & L score: Kellgren and Lawrence score

## Materials and methods

### Ethics considerations for the present study

Experimental procedures in this study were conducted in vitro and did not involve the use of animal models or samples, patient samples or data, or recruitment of human subjects. There-fore, research conducted as part of this study posed minimal risk and fits one of the exempt review categories as defined by institutional review board (IRB) regulations at Mohammed Bin Rashid University (MBRU). Further clarification and information can be obtained from the MBRU IRB at irb@mbru.ac.ae.

The study involved only in vitro experiments with commercially available cell lines that do not involve the use of human subjects. In addition, we would like to emphasize that no minors were included in this study, and therefore parental or guardian consent was not necessary. Experiments conducted in this study did not involve any form of direct or indirect interaction with human subjects. It is important to note that our study did not involve the use of medical records or archived samples.

In summary, our study involved in vitro experiments using commercially purchased cell lines and did not involve human subjects. Therefore, no participant consent was required, and the need for consent was waived by the ethics committee.

## Cell line

Human bone marrow mesenchymal stem cell line (BMSC) was purchased from AddexBio Technologies, CA, USA. Upon receiving the cells in liquid nitrogen, they were immediately washed with phosphate buffered saline (PBS, Gibco, USA) to ensure that the cells were free from the cryoprotectant Dimethyl Sulfoxide (DMSO). The washed cells were then cultured in mesenchymal stem cell growth media (MSCM, AddexBio Technologies, CA, USA) supplemented with 10% Fetal Bovine Serum (FBS, Himedia, USA) and 1% penicillin streptomycin antibiotic solution (Himedia, USA). After the addition of MSCM, the cells were kept in an incubator supplied at 37˚C temperature and 5% CO2. The cell culture medium was replaced with fresh medium at an interval of three days, and their confluency appraised. Once the cells achieved 90% confluency, the proteolytic enzyme trypsin was added to dislodge the adherent cells from the culture flask. Trypsinization was done using trypsin/EDTA solution (Himedia, USA). The detached cells were sub-cultured for further growth in MSCM.

## Reagents

Complete MesenCult™-ACF chondrogenic differentiation medium was purchased from Stem cell Technologies$^{TM}$ (BC, Canada). Vitamin D (Calcitriol) (1,25-dihydroxyVitamin D3 (1,25 (OH)2D3)), lipopolysaccharide (LPS) and curcumin were purchased from Thermofisher Scientific (CA, USA). We used calcitriol in all our experiments/assays as it is the most active form of Vitamin D [57]. Toluidine blue ($C_{15}H_{16}N_3S^+$) and Alcian blue ($C_{56}H_{68}C_{14}CuN_{16}S_4$) stains were purchased from abcam, Boston, US. ELISA kits were purchased from Assay Genie, Ireland.

## Cell culture

After five passages of subculture (Passage 5), BMSCs were used for chondrogenic differentiation. For chondrogenic differentiation, $2x10^6$ BMSCs (estimated using automated cell counter (DeNovix, Cell Drop, USA) were pelleted in a 15 mL polypropylene tubes after centrifugation at 1000 rpm for 10 minutes. Two mL of complete MesenCult™-ACF chondrogenic differentiation medium was added to the cell pellet, from which 0.5 mL of the cell suspension was added to each 15 mL polypropylene tube (four in total). Each of these tubes was centrifuged at 1000 rpm for 10 minutes at 25˚C using a centrifuge (Thermofisher Scientific, USA). Caps were gently loosened, and the tubes were incubated at 37˚C in the presence of 5% $CO_2$, to ensure no alterations in the pH of the cell culture media. Following three days of incubation, 0.5 mL of complete MesenCult™-ACF chondrogenic differentiation medium was added to the tubes, followed by incubation at 37˚C in the presence of 5% $CO_2$ over three days. After incubation, the cell culture medium was carefully aspirated from each tube, and 0.5 mL of fresh MesenCult™-ACF chondrogenic differentiation medium was added. The tubes were further incubated at 37˚C in the presence of 5% $CO_2$ for 21 days, with intermittent replacement (at three days interval) of cell culture medium with fresh MesenCult™-ACF chondrogenic differentiation medium. After each change of cell culture medium, the differentiating cell pellets were gently "flicked" to ensure that pellets did not adhere to the tube surfaces. After 21 days of incubation, the cell pellets, now at the stage of chondrogenic differentiation to pre-chondrocytes, were further incubated for seven days.

## Cryopreservation of differentiated chondrocytes

Chondrocytes, intended for cryopreservation, were centrifuged at 1000 rpm for 10 minutes. The cells were then resuspended in the freezing medium (Dulbecco's Modified Eagle's Media

with 5% DMSO) and transferred into suitable sized cryovials. After being kept at -80˚C overnight, the pellets were then stored in a liquid nitrogen tank for long-term preservation.

## Preparation of histological section from cell pellet

The pellet was transferred to a histology cassette followed by fixation in 10% neutral-buffered formalin for 24 hours. The cassette, along with the pellet, was transferred to 70% ethanol and then subjected to dehydration using a series of graded ethanol solutions (25%, 50%, 75%, 90%, 95%, and 100%, each for 3 minutes). Clarification steps were carried out in xylene and embedded in paraffin (Surgipath, USA) following standard embedding protocol. Cell pellets containing chondrocytes were sectioned (6 μm) using a microtome (Leica RM2255, Germany). Sections were mounted on slides. The slide sections were rehydrated by a decreasing series of alcohol (100%, 95%, 90%, 75%, 50%, and 25%, each for three minutes) following deparaffinization with three changes of xylene. Finally, the sections were rinsed with deionized water for five minutes.

## Alcian blue staining

Alcian blue staining was done according to the method of Lu et al., 2017 with minor modifications [58]. The solution for Alcian blue was prepared by dissolving 1 gm of Alcian blue in 3% acetic acid and adjusting the pH to 2.5. The pre-prepared slides, which contained 6 μm sections of the pellet, were placed in a 3% acetic acid solution for 3 minutes, followed by immersion in a 1% solution of Alcian blue for 30 minutes. Next, the slides were washed with running water for one minute and then dehydrated using a series of graded ethanol concentrations (25%, 50%, 75%, 90%, 95%, and 100%, each for 3 minutes).The slides were allowed for clarification in xylene and mounted in mounting solution (Permount® (Fisher)). The slides were then observed using a microscope.

## Toluidine blue staining

Toluidine blue staining was done according to the method of Bergholt et al., 2019 with minor modifications [59]. The solution for toluidine blue was prepared by dissolving 0.1 gm of toluidine blue in 100 mL of deionized water and adjusting the pH to 4. After the preparation of slides (as described above), they were stained with toluidine blue and left for two minutes at room temperature. The slides were then rinsed carefully with water for one minute and dehydrated using a series of graded ethanol concentrations (25%, 50%, 75%, 90%, 95%, and 100%, each for three minutes). The slides were allowed for clarification in xylene and mounted in mounting solution (Permount® (Fisher)). The slides were then observed using a microscope.

## Microscopic analysis

Olympus BX63 microscope (Japan) was used for analysis of staining. The microscope is equipped with Olympus DP80 camera (resolution of 4080 x 3072 pixels and pixel size of 6.45 X 6.45 micrometer) with Cell Sens Dimension 2.3 software (version 18987). Images were acquired with 20X (numerical aperture 0.45; 1920 x 1200 pixels; 465.079 nm/ pixel resolution in both X and Y axis) and 40X (numerical aperture 0.6; 1920 x 1200 pixels; 232.54 nm/pixel resolution in both X and Y axis) objective lenses.

## Assessment of cytotoxicity through MTT assay

Our examination of the cytotoxicity of LPS, Vitamin D, and Curcumin was conducted using the MTT (3-(4,5-Dimethylthiazol-2-yl)-2,5-diphenyltetrazolium bromide) assay. This highly

recognized colorimetric assay is extensively employed to gauge cell metabolic activity, serving as an indirect measure of cell viability and cytotoxicity.

The principle behind the MTT assay hinges on the activity of mitochondria within viable cells. Active mitochondria reduce MTT to an insoluble formazan, a purple compound. The quantity of formazan produced, measured by optical density, corresponds directly with the number of metabolically active cells.

To investigate the cytotoxic effects of various doses of LPS (0.1 μg/mL, 1 μg/mL, 10 μg/mL, 20 μg/mL, and 40 μg/mL), Vitamin D (0.01 μM, 0.02 μM, 0.12 μM, 0.25 μM), and Curcumin (10 μM, 25 μM, 50 μM, 75 μM, 100 μM), we cultured cells in a 96-well plate. After 24 hours of treatment, we added 20 μL of an MTT solution (prepared at a concentration of 5 mg/mL in PBS) to each well and incubated the plate at 37° C with 5% $CO_2$ for 4 hours.

Following this, we discarded the supernatant and dissolved the formed formazan crystals in 1 mL of 100% dimethyl sulfoxide (DMSO), allowing a further incubation period of 30 minutes at 37° C, 5% CO2. This solubilizes the formazan crystals, readying them for measurement. We then measured the optical density (OD) of the resultant solution at a wavelength of 570 nm using a Hidex plate reader. The optical density reflects the quantity of formazan and, by proxy, the number of viable cells. Consequently, relative cell viability was determined as the percentage of the live cell density, calculated with the formula:

$$\frac{100 \times (OD_{sample} - OD_{blank})}{OD_{control}}$$

MTT in DMSO solution was used as the blank [60]. Thus, by providing a clear, accurate, and reproducible measure of cell viability, the MTT assay allowed us to effectively assess the cytotoxicity of these specific doses of LPS, Vitamin D, and Curcumin. This helped further our understanding of their potential therapeutic implications in the management of osteoarthritis.

## Creation and evaluation of pro-inflammation model

LPS was used to stimulate inflammation in the differentiated chondrocytes to create the pro-inflammation model. LPS was administered at 10 μg/mL concentration to the differentiated chondrocytes for induction of inflammation. We did not increase the dose of LPS to augment inflammatory response in the chondrocytes as higher doses of LPS can overstimulate cells, leading to excessive inflammation and potentially cell death, which can make it difficult to interpret experimental results [61]. The cells were incubated in an incubator at 37°C and 5% $CO_2$ concentration for one day.

## Cytokine expression study

Expression of TNF α, IL 6, IL 8 and IFN γ was assessed in the pro-inflammatory chondrocyte model created in this study (*refer below for details*) with/without Vitamin D and curcumin. Both Vitamin D (in the form of calcitriol) and curcumin are known to be relatively insoluble in aqueous buffers. To overcome this, we dissolved these bioceuticals in 0.01% DMSO. Prior to this, we performed viability assays on chondrocytes and confirmed that 0.01% DMSO had no deleterious effects on cell viability (Refer to Fig 3 (*control (0) in insets A-C*)). Additionally, we evaluated the effect of the buffer containing DMSO on the expression of GAPDH protein in chondrocytes (*refer to lane 1 in Fig 4*) and found no detrimental effects of DMSO on protein expression. Chondrocyte supernatants were collected following centrifugation at 5000 rpm for 10 minutes. Concentrations of different cytokines were determined through the use of commercially available enzyme linked immunosorbent assay (ELISA) kit. The assay was executed as per the manufacturer's guideline on ELISA kits (ELISAGenie, Ireland). The plates were

precoated with human cytokines. Levels of cytokines were measured using a microplate reader (Hidex, Japan) at 450 nm with a reference wavelength of 620 nm. All experiments were conducted independently for at least three times unless otherwise mentioned. All data were reported as mean ± standard deviation. Data was analyzed using GraphPad prism version 9.5.1.

## Western blotting

Total proteins were extracted with 0.5% SDS (Invitrogen, USA) supplemented with protease inhibitors. The protein level was determined using the bicinchoninic acid protein assay, and 20 μg of the protein electrophoresed on a discontinuous 10% SDS gel polyacrylamide. The proteins were transferred electrophoretically onto a nitrocellulose membrane (Bio-Rad Laboratories, Canada) for 1 h at 4˚C. The membranes were incubated for 1h at 4˚C with 3% bovine serum albumin (BSA) in Tris–buffered saline (Pierce, USA). The membranes were then washed once with tween tris buffered saline (TTBS) (Tris 20mM, NaCl 150mM, pH 7.5, and 0.1% Tween 20) for 10 min and incubated in SuperBlock 1:10 in TTBS, supplemented with the mouse anti-human PAR-2 antibody (1:500; USA) overnight at 4˚C. The membranes were washed with TTBS and incubated for 1 h at room temperature with the second antibody (1:2000; anti-mouse IgG horseradish peroxidase conjugated; Pierce) and washed again with TTBS. Detection was performed by chemiluminescence using the Super Signal ULTRA chemiluminescent substrate (Pierce) and exposure to Kodak Biomax photographic film (GE Healthcare, Canada). The band intensity was measured by densitometry using image J Software, and data were expressed as arbitrary units.

## Flow cytometry studies

Cells were washed with 1X PBS, detached with the trypsin/EDTA solution (Himedia, USA) at 37˚C, and centrifuged at 500 g for 5 min at 4˚C. The cells were re-suspended in 1X PBS and a 500 μL suspension prepared, with a concentration of $1X10^6$ cells/mL. The suspension was incubated for 30 min at room temperature and divided into tubes. For the determination of RANK and RANKL, one served as negative control to which a mouse IgG (15 μg/mL; Chemicon International, USA) for RANKL and a mouse IgG coupled to phycoerythrin (IgG-PE: 20 μg/ml; R&D Systems) for RANK was added, and the other was labelled with either a mouse anti-human RANKL antibody (15 μg/ml; R&D Systems) or a mouse anti-human RANK-PE (20 μg/ml; Abcam) for 30 min at 4˚C. For RANKL, after this period, cells were washed, and a goat anti-mouse FITC-conjugated secondary antibody (7.5 μg/ml; R&D Systems) was added for another 30 min at 4˚C. Cells were then re-suspended in PBS and analyzed by using a flow cytometry apparatus (Facscalibur; BD Biosciences, Canada). The control sample was used to determine the background fluorescence and compared with that of the sample incubated with the specific antibody.

## Targeted regulation of PAR-2 expression via siRNA transfection and evaluation of LPS dose-dependent response

To evaluate the potential functional role of PAR-2 in mediating TNF α expression, we employed the use of small interfering RNAs (siRNAs) to specifically target and suppress PAR-2 expression in chondrocytes. The use of siRNA is a robust and reliable methodology for the targeted knockdown of specific genes, allowing us to assess the implications of PAR-2 reduction in our model.

The siRNA was prepared by combining it with Opti-MEM solution in a 1:4 ratio in sterile falcon tubes. Separately, Oligofectamine was diluted in Opti-MEM solution at a 1:3 ratio. We then added 50 μL of the Oligofectamine mixture to the pre-prepared siRNA solution. This combined solution was allowed to incubate for 15 minutes at room temperature to facilitate

the formation of siRNA-Oligofectamine complexes. The chondrocytes, housed in a 100mm dish, were washed thrice with serum-free DMEM to ensure minimal interference during transfection. Following this, 3 mL of DMEM was amalgamated with the siRNA-Oligofectamine complex, creating the final transfection mixture. This mixture was then added to the dish containing the chondrocytes, maintaining a final siRNA concentration of 0.1 μM. The cells were subsequently incubated for 5 hours at 37˚C, allowing for siRNA uptake and gene knockdown. After this incubation period, we replenished the media with complete DMEM supplemented with 10% FBS and continued the incubation at 37˚C for an additional 24 hours. This was done to allow sufficient time for the effects of PAR-2 suppression to manifest. At the end of this period, we isolated proteins from the cells and performed western blotting to confirm the successful knockdown of PAR-2 and investigate any consequential changes in TNF α expression.

Furthermore, we also examined the effects of increased LPS concentrations on PAR-2 expression. Specifically, we studied concentrations of 20 μg/mL and 40 μg/mL, in addition to the initially used 10 μg/mL that induced PAR-2 expression, to study the dose-dependent influence of LPS on PAR-2 expression in the proinflammatory chondrocyte model and any consequential changes in TNF α expression.

## Results

### Creation of the proinflammatory chondrocyte mode

The workflow demonstrating the creation of the proinflammatory chondrocyte model is shown in Fig 1.

The chondrocytes were effectively differentiated from human BMSCs after cultivation in chondrogenic differentiation media, as demonstrated in Fig 2A. The quality of the differentiated chondrocytes was evaluated histologically using two different dyes: Alcian blue and Toluidine blue. The successful chondrogenic differentiation of BMSCs led to the expression of proteoglycan (aggrecan), type II collagen, and type X collagen after 21 days, which was verified through Alcian blue staining (Fig 2B). Further evaluation was carried out using Toluidine blue staining, which specifically binds to the sulfate groups of proteo-glycans. The well-differentiated chondrocytes were observed to stain purple due to the presence of proteoglycans in the extracellular matrix, as seen in the present study (Fig 2C).

### Assessment of cytotoxicity of LPS, Vitamin D, and curcumin

In our exploration of the cytotoxicity of LPS, Vitamin D, and curcumin at varying concentrations, we employed the MTT assay, a reliable and robust method for measuring cellular metabolic activity and viability. As depicted in Fig 3, our results demonstrated a remarkable lack of cytotoxic effects across all the doses evaluated for each compound.

Specifically, Fig 3A illustrates the effects of different concentrations of LPS (0.1 μg/mL, 1 μg/mL, 10 μg/mL, 20 μg/mL, and 40 μg/mL) on the live cell density. Regardless of the dose, LPS did not exhibit cytotoxicity, suggesting that these dosages are safe for use in our model of BMSC-derived chondrocytes to create proinflammatory chondrocyte model.

Similarly, Vitamin D showed no cytotoxic effects at any of the evaluated doses (0.01 μM, 0.02 μM, 0.12 μM, 0.25 μM) as presented in Fig 3B. This indicates that Vitamin D, at these concentrations, retains its biocompatibility, further supporting its potential utility in therapeutic applications for OA.

Fig 3C showcases the effects of various concentrations of Curcumin (10 μM, 25 μM, 50 μM, 75 μM, 100 μM) on the cell density. Mirroring the results of LPS and Vitamin D, curcumin, across all tested doses, did not induce cytotoxic effects. This underscores the biocompatibility

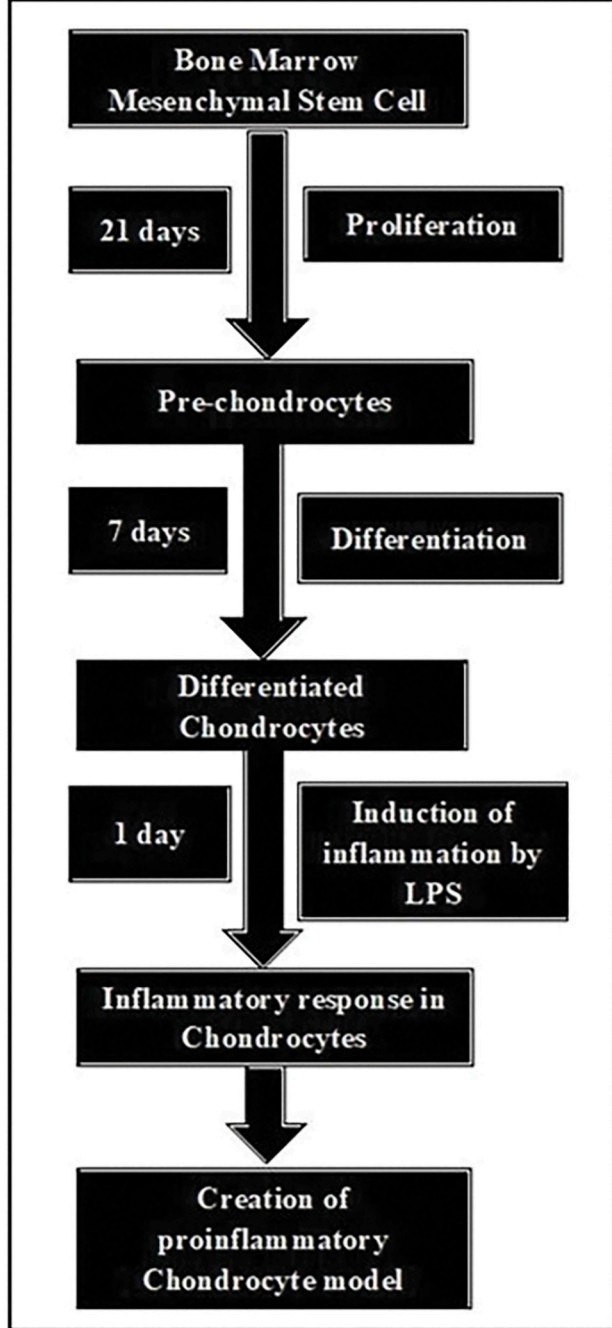

**Fig 1. Creation of the proinflammatory chondrocyte model.** (Note: *Inflammation was assessed by assessing the levels of TNF α, IL 6, IL 8 and IFN γ in LPS treated chondrocytes versus LPS naive chondrocytes (control)*).

of curcumin, even at higher concentrations, strengthening its candidacy as a potential natural treatment for OA.

Taken together, these findings contribute valuable insights into the safety profiles of LPS, Vitamin D, and curcumin at the administered doses. This absence of cytotoxicity observed in our study affirms their potential for use in therapeutic interventions for OA without compromising cellular viability.

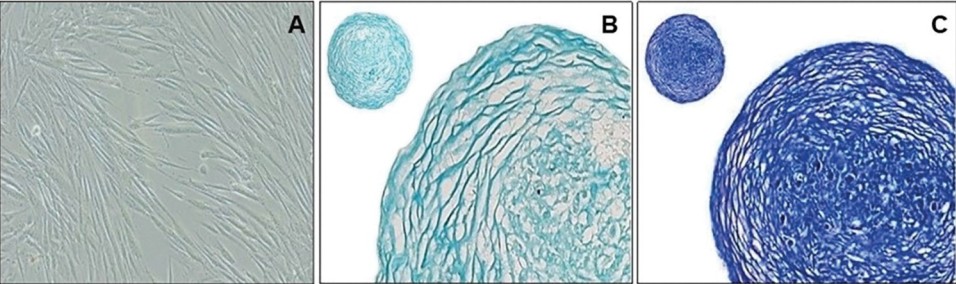

**Fig 2. Qualitative analysis of differentiated chondrocytes.** (a) Differentiated chondrocytes obtained from BMSC after 28 days; Differentiated Chondrocytes stained with (b) Alcian blue (c) Toluidine blue.

The reason we selected these cytokines as molecular markers in our proinflammatory model was because of the critical role of these cytokines in the inflammatory responses that occur in OA [62] in PAR-2 mediated signaling. TNF α has been extensively studied in the context of joint inflammation in patients with OA, where it was identified as a key target for therapy [62]. This led to the development of several therapeutic strategies aimed at targeting TNF α, such as etanercept, trastuzumab, adalimumab, and infliximab to treat patients with OA [15–17]. In addition to TNF α, IL 6, IL 8, and IFN γ are also essential cytokines that play a significant role in the pathogenesis of OA. IL 6 is a pro-inflammatory cytokine that contributes to the production of MMPs and proteoglycans, which are important components of cartilage [63]. IL 8, another pro-inflammatory cytokine, is involved in recruiting leukocytes to the joint,

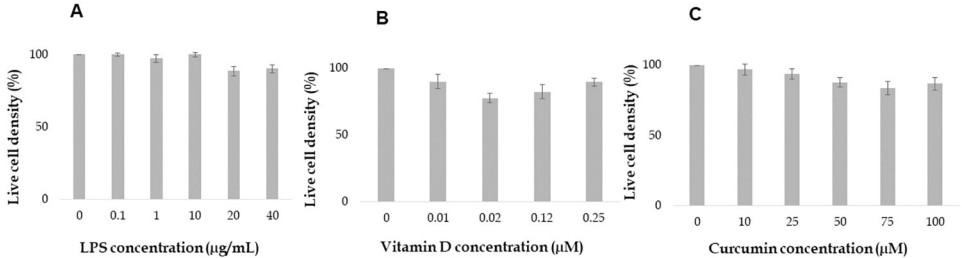

**Fig 3. Cytotoxicity assessment of LPS, Vitamin D, and curcumin using MTT assay.** This figure visually represents the cytotoxicity evaluation of LPS, Vitamin D, and Curcumin at varying doses on BMSC-derived chondrocytes using the MTT assay. **(A) LPS cytotoxicity**: The graph showcases the live cell density after treatment with different concentrations of LPS (0.1 μg/mL, 1 μg/mL, 10 μg/mL, 20 μg/mL, and 40 μg/mL). (*Note*: *The results demonstrate no observable cytotoxic effect at any of the tested LPS doses.*); **(B) Vitamin D cytotoxicity**: The plot represents the cell density after exposure to various doses of Vitamin D (0.01 μM, 0.02 μM, 0.12 μM, 0.25 μM). (*Note*: *There is no evident cytotoxicity*, *affirming the biocompatibility of Vitamin D at these concentrations.*); **(C) Curcumin cytotoxicity:** The graph displays the live cell density following treatment with differing concentrations of curcumin (10 μM, 25 μM, 50 μM, 75 μM, 100 μM). (*Note*: The data show a lack of cytotoxic effects across all evaluated curcumin doses.) *The overall absence of cytotoxicity across all tested doses of LPS*, *Vitamin D*, *and Curcumin*, *demonstrates their safety for use in therapeutic applications for OA.*After successfully generating high-quality chondrocytes from BMSCs, we aimed to create a "pro-inflammation model" by inducing inflammation in the chondrocytes. We accomplished this by usingLPS, a biomolecule commonly found in the outer membrane of Gram-negative bacteria, which is well-known for being a potent stimulus for pathological dysfunctions. In septicemia, circulating LPS acts as a pathogen-associated molecular pattern (PAMP) and triggers the innate immune system, resulting in local or systemic inflammatory responses [61]. It has been established that LPS can also stimulate non-immune cells and elicit severe inflammatory reactions via dedicated receptors. For instance, the innate LPS-pattern recognition receptor Toll-like receptor 4 (TLR4) is expressed extensively in cardiomyocytes, leading to severe LPS-induced inflammation in cardiomyocytes, even in the absence of immune cells [10]. Similarly, TLR4 is also expressed in chondrocytes [13]. Based on these findings, we hypothesized that LPS would be able to initiate proinflammatory responses in differentiated chondrocytes without the participation of immune cells. To evaluate the extent of inflammation, we measured the levels of Tumor Necrosis Factor alpha (TNF α), Interleukin 6 (IL 6), Interleukin 8 (IL 8), and Interferon gamma (IFN-γ) in LPS-treated chondrocytes and compared them to those in LPS-naive chondrocytes (the control group) (Fig 4A to 4D).

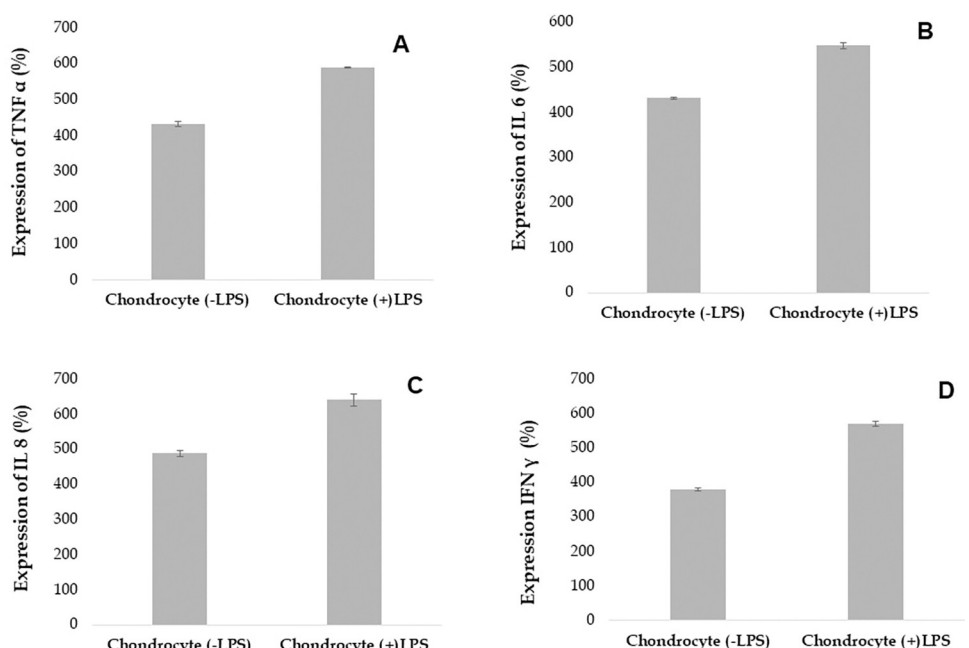

**Fig 4. Assessment of proinflammatory markers in the chondrocyte model of inflammation.** The X-axis represents LPS-naive chondrocytes (control) control and LPS induced chondrocytes whereas the Y-axis represents expression of the cytokines in pg/mL. (Note: *In the presence of LPS the expression of all the cytokines were upregulated*).

leading to cartilage destruction [64]. The levels of IFN γ in OA have been reported to be both increased and decreased in different studies. IFN γ is a type of cytokine with both pro-inflammatory and anti-inflammatory properties [65], and its role in OA is complex and not fully understood. In some studies, elevated levels of IFN γ have been observed in the synovial fluid and serum of patients with OA, suggesting a pro-inflammatory role for this cytokine in the disease [66]. On the other hand, other studies have reported decreased levels of IFN γ in OA, indicating that this cytokine may play a more anti-inflammatory role in the disease [67]. To assess the levels of TNF α, IL 6, IL 8, and IFN-γ, differentiated chondrocytes were incubated for 24 hours after LPS treatment, and the levels were measured using an ELISA. The results showed that LPS-treated chondrocytes expressed 34% higher levels of TNF α compared to the LPS-naive chondrocytes (the control group, which was treated with phosphate buffered saline (PBS) instead of LPS). Whereas the chondrocytes treated with LPS expressed 42% higher IFN-γ level than the control group. In case of IL 6 and IL 8, LPS- treated chondrocytes expressed 26% and 32% higher interleukins as compared to the control group in presence of PBS respectively as shown in Fig 4A to 4D.

## Study of PAR-2 expression in the presence/absence of Vitamin D and curcumin in the proinflammatory chondrocyte model

Protease-activated receptor 2 (PAR-2) is a type of cell-surface receptor that is involved in the regulation of inflammation and pain. In the context of OA, PAR-2 has been shown to play a role in the regulation of cartilage degradation and pain [40]. Studies have shown that PAR-2 is expressed in chondrocytes and synoviocytes, which are key cell types involved in the development of OA. Activation of PAR-2 in these cells has been shown to lead to increased production of pro-inflammatory cytokines MMPs [68]. This increased activity of pro-inflammatory cytokines and MMPs contributes to the progression of OA and the associated pain. In addition to

its pro-inflammatory effects, PAR-2 has also been shown to play a role in pain regulation in OA [69]. Activation of PAR-2 in sensory nerve endings has been shown to lead to the release of pain-causing substances, such as substance P and calcitonin gene-related peptide (CGRP), which can contribute to the development of joint pain in OA. As there is limited information on the subject, we investigated the impact of Vitamin D and curcumin on PAR-2 expression using Western blotting. From the experiment it was observed that PAR 2 is expressed more in LPS induced chondrocytes than control cells (LPS naive). Vitamin D at 0.12 and 0.25 μM concentration reduced the expression of PAR 2 in LPS induced chondrocytes (Fig 5A). Analysis of the densitometric data indicated that chondrocytes induced with LPS expressed approximately 20% more PAR-2 compared to LPS-naive chondrocytes. However, treatment of the inflammatory chondrocytes with a concentration of 0.12 μM of Vitamin D led to a reduction of approximately 31% in the expression of PAR-2, as shown in Fig 5B. Additionally, treatment with 0.25 μM of Vitamin D resulted in a further reduction of PAR-2 expression to 70%, as also depicted in Fig 5B. Curcumin was found to exhibit anti-inflammatory effects by suppressing the expression of PAR-2. Induction of LPS led to increased expression of PAR-2 in chondrocytes, which was mitigated by treatment with 50 and 100 μM concentrations of curcumin, as shown in Fig 5C. Densitometric analysis revealed that treatment with 50 μM and 100 μM doses of curcumin resulted in a reduction of approximately 20% and 70%, respectively, in the expression of PAR-2, as depicted in Fig 5D. Concurrently, we evaluated the expression of the reference gene glyceraldehyde-3-phosphate dehydrogenase (GAPDH) in LPS-treated and LPS-naïve chondrocytes with and without Vitamin D and curcumin treatment. Our results, as depicted in Fig 5E and 5F, reveal that neither bioceutical has any significant impact on the expression of GAPDH. Therefore, it is evident that the anti-inflammatory effects of Vitamin D and curcumin are specifically related to the reduction of PAR 2 expression.

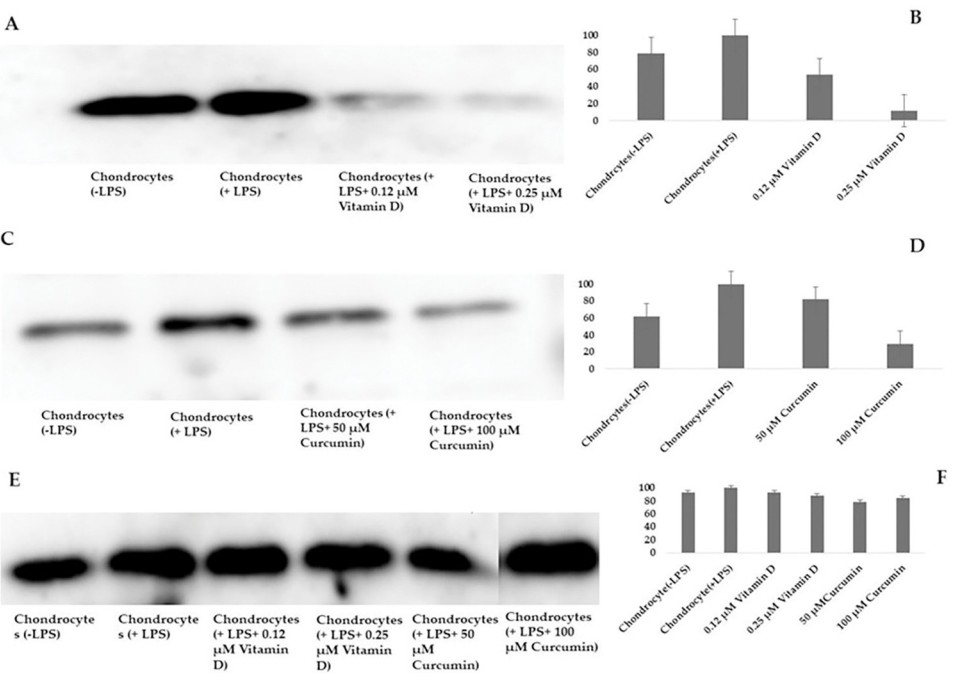

**Fig 5. Western blot analysis of chondrocytes in +/- of LPS after treatment with Vitamin D and curcumin.**

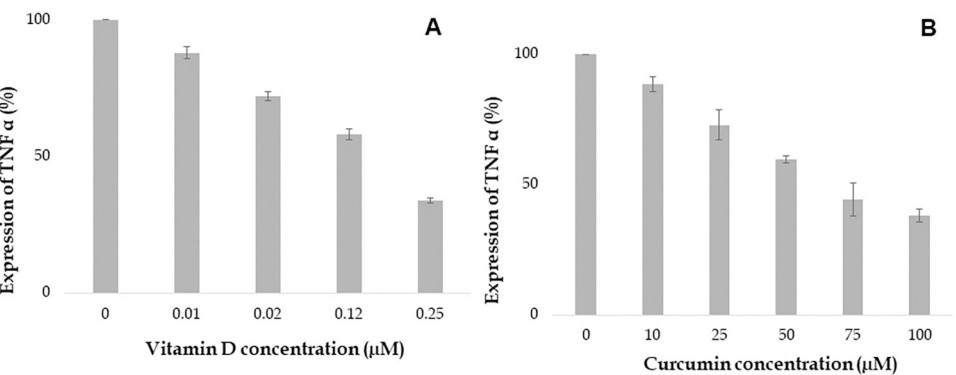

**Fig 6. Increasing concentration of Vitamin D and curcumin attenuates the expression of TNF α (%).** Values at x-axis represents the concentrations in μM and y-axis represents expression of TNF α in %.

## Assessment of anti-inflammatory properties of Vitamin D and curcumin in the proinflammatory chondrocyte model

Vitamin D attenuates expression of cytokines such as TNF α, IL 6, IL 8 and IFN-γ which are involved in the inflammatory response. Our study aimed to investigate the impact of various concentrations of Vitamin D on cytokine expression in the human proinflammatory chondrocyte model. The findings demonstrated that Vitamin D reduced the expression of TNF α, IL 6, IL 8 and IFN-γ in a dose-dependent manner as shown in Figs 6 to 9. Specifically, higher concentrations of 0.12 and 0.25 μM, Vitamin D effectively reduced TNF α expression, as depicted in Fig 6A. Furthermore, IL 6 and IL 8 expressions were found to gradually decrease at different doses of Vitamin D, ranging from 0.01 μM to 0.25 μM, as observed in Figs 7A and 8A respectively. Additionally, the expression of IFN-γ decreased at concentrations of 0.02 μM, followed by 0.12 and 0.25 μM, as depicted in Fig 9A. Notably, Vitamin D exhibited greater potency and efficacy at 0.12 and 0.25 μM concentrations.

Curcumin was found to reduce the expression of inflammatory cytokines such as TNF α, IL 6, IL 8, and IFN-γ in a dose-dependent manner. Treatment with 25 μM curcumin decreased the expression of TNF α, as shown in Fig 6B. The most significant decrease in TNF α expression was observed at 100 μM concentration of curcumin. Similarly, treatment with various concentrations of curcumin (ranging from 25 to 100 μM) resulted in a gradual reduction in the expression of IL 6 and IL 8, as observed in Fig 7B and 8B, respectively. Furthermore, the expression of IFN-γ was also decreased at several concentrations of curcumin (Fig 9B). Notably, curcumin exhibited greater potency and efficacy at concentrations of 50, 75, and 100 μM to the human chondrocytes (Figs 6B, 7B, 8B, and 9B).

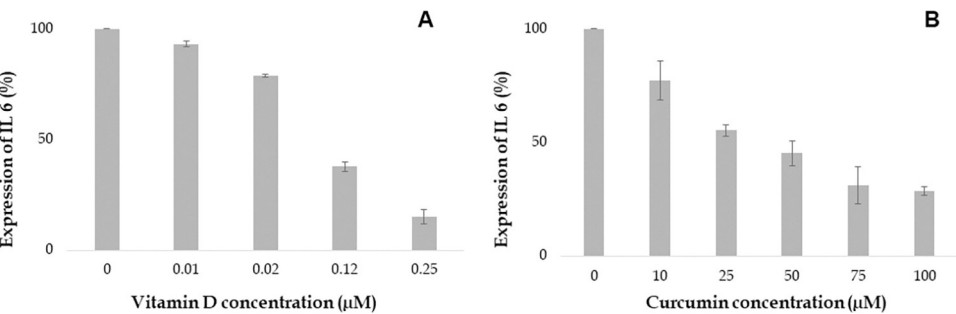

**Fig 7. Increasing concentration of Vitamin D and curcumin attenuates the expression of IL 6 (%).** Values at x- axis represents the concentrations in μM and y-axis represents expression of IL 6 in %.

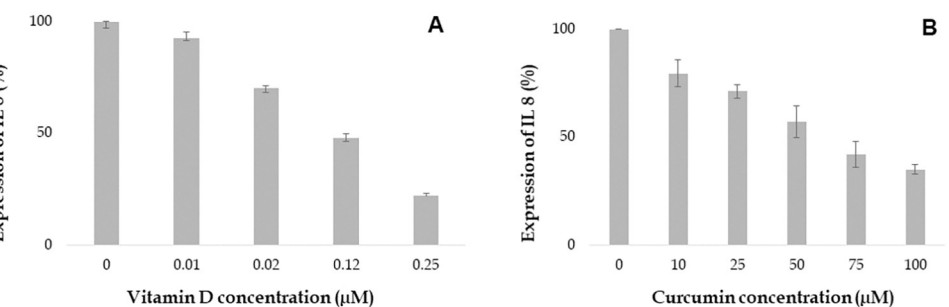

**Fig 8. Increasing concentration of Vitamin D and curcumin attenuates the expression of IL 8 (%).** Values at x- axis represents the concentrations in μM and y-axis represents expression of IL 8 in %.

## Flow cytometric assessment of expression of RANK and RANKL in in the presence/absence of Vitamin D and curcumin in the proinflammatory chondrocyte model

Receptor activator of nuclear factor kappa-B (RANK) and its ligand (RANKL) play an important role in the development of OA. RANK is a transmembrane receptor protein that is expressed on the surface of various cell types, including chondrocytes, and is activated by binding with RANKL. In the healthy joint, RANKL signals regulate the differentiation, activation, and survival of osteoclasts, which are responsible for breaking down and remodeling bone. However, in OA, there is an imbalance between RANKL and its decoy receptor, osteoprotegerin (OPG), leading to an excessive activation of osteoclasts and an increased rate of bone and cartilage degradation. This excessive degradation results in the formation of osteophytes, which are bony protrusions that can further contribute to the progression of OA. Additionally, studies have shown that RANKL and its signaling pathway also play a role in chondrocyte activation and matrix degradation. RANKL can induce the production of matrix-degrading enzymes and pro-inflammatory cytokines by chondrocytes, contributing to the progression of OA. Overall, the RANKL-RANK pathway has been shown to play a critical role in the pathogenesis of OA, and targeting this pathway has been proposed as a potential therapeutic strategy for OA. In line we assessed the effect of both Vitamin D and curcumin on RANK and RANKL expression in the proinflammatory chondrocyte model (Fig 10). Both RANK and RANKL are not expressed in control chondrocytes (Fig 10A and 10F). The cells after induction with LPS expressed RANK (Fig 10B and 10G). However, RANK expression is reduced after treating the inflammatory chondrocytes at 0.25 μM Vitamin D and 100 μM curcumin concentration respectively (Fig 10C and 10H). RANKL expression was also increased in the pro-inflammatory chondrocyte model (Fig 10D and 10I) which was reduced after 0.25 μM Vitamin D and 100 μM curcumin treatment (Fig 10E and 10J).

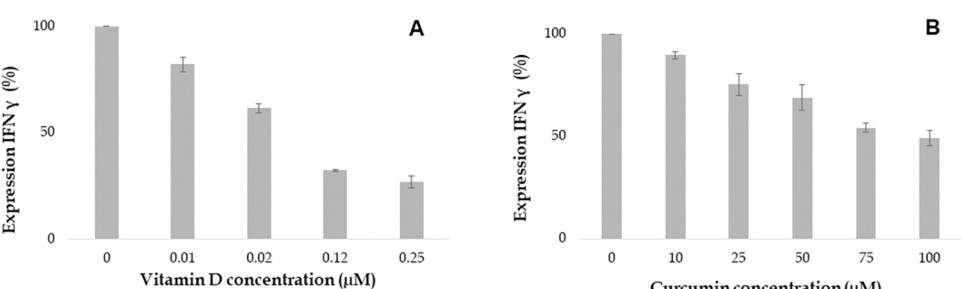

**Fig 9. Increasing concentration of Vitamin D and curcumin attenuates the expression of IFN γ (%).** Values at x-axis represents the concentrations in μM and y-axis represents expression of IFN γ in %.

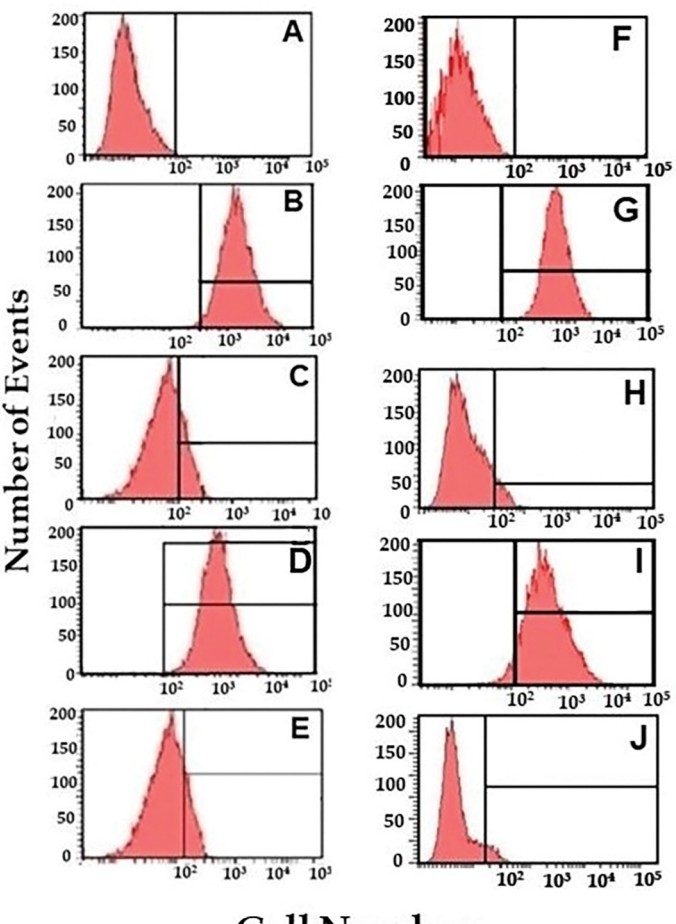

**Fig 10. The expression of RANK and RANKL in control cells and inflammatory chondrocytes treated with Vitamin D and curcumin.** Panels A and F show control cells with no expression of RANK and RANKL, while panel B and D show inflammatory chondrocytes expressing RANK. Panel C demonstrates a reduction in RANK expression after treatment with 0.25 μM of Vitamin D concentration, and panel E shows a reduction in RANKL expression after the same concentration of Vitamin D treatment. Panel G depicts a reduction in RANK expression after treatment with 100 μM of curcumin concentration, while panel J shows a reduction in RANKL expression after the same concentration of curcumin treatment.

## siRNA mediated silencing of PAR-2 expression in chondrocytes under LPS stimulation

Upon siRNA-mediated transfection, we found a significant suppression of PAR-2 expression, even in the presence of varied concentrations of LPS, including 0.1 μg/mL, 1 μg/mL, and 10 μg/mL—the latter being the concentration employed for establishing the pro-inflammatory chondrocyte model (as evident in Fig 11A and 11B). Notably, even the escalated doses of LPS (20 μg/mL and 40 μg/mL) failed to reinstate the PAR-2 expression (Fig 11A and 11B). Crucially, this specific downregulation of PAR-2 via siRNA did not appear to disrupt the expression of the control gene, GAPDH, irrespective of the LPS concentration gradient (Fig 11C and 11D).

   These observations underscore the potent and precise role of siRNA in gene silencing, effectively curtailing PAR-2 expression under all tested LPS conditions. The unchanged GAPDH expression substantiates the selectivity of the siRNA and reinforces its reliability as a gene manipulation tool. Moreover, the fact that increased LPS concentrations failed to revive

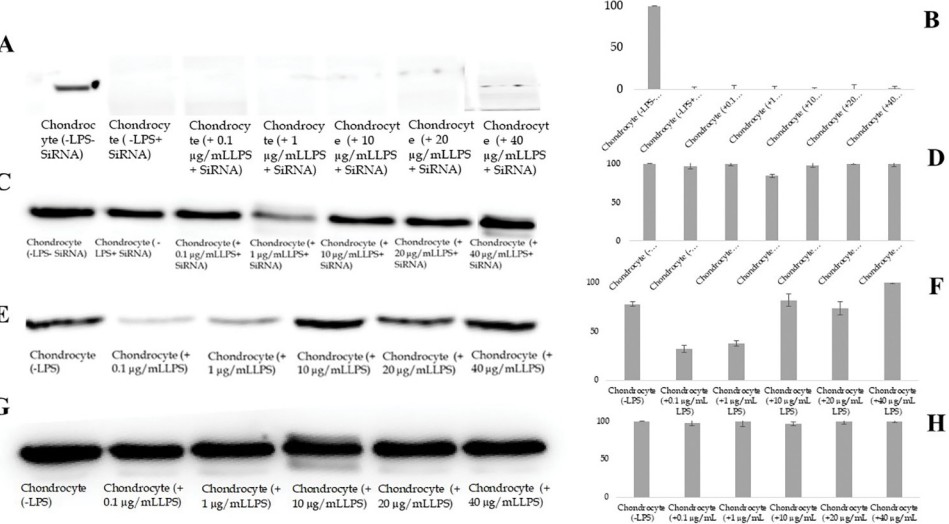

**Fig 11. Impact of siRNA-mediated PAR-2 silencing on chondrocytes in response to varying concentrations of LPS.** (A-B) PAR-2 expression levels in siRNA-transfected chondrocytes post LPS treatment, highlighting the persistent downregulation of PAR-2 even in the presence of increased concentrations of LPS. (C-D) Expression of the control gene, GAPDH, in siRNA-transfected chondrocytes, establishing the selective action of siRNA-mediated PAR-2 silencing and no broad cytotoxic effects. (E-F) Expression of PAR-2 in chondrocytes exposed to escalating doses of LPS, showcasing the dose-dependent upregulation of PAR-2 in the absence of siRNA. (G-H) GAPDH expression in non-transfected chondrocytes subjected to increasing LPS concentrations, verifying the unaffected control gene expression. Each bar represents the mean ± SD.

PAR-2 expression reaffirms the robustness of our siRNA-based silencing approach. Lastly, the integrity of GAPDH expression across all conditions substantiates that our experimental manipulations did not have broad cytotoxic effects (confirmed earlier through MTT assay), but rather, were limited to the targeted suppression of PAR-2.

In contrast, in the absence of siRNA transfection, we noted a dose-dependent upregulation of PAR-2 expression with increasing LPS concentrations, validating the role of LPS in augmenting PAR-2 expression (Fig 11E and 11F). However, this change did not affect the expression of the control gene, GAPDH, which remained consistent despite varying LPS concentrations (Fig 11G and 11H).

Collectively, these observations elucidate the critical role of PAR-2 in LPS-mediated inflammation in chondrocytes. They also highlight the effectiveness and precision of siRNA-mediated gene silencing, even in the face of heightened proinflammatory stimuli. Furthermore, the data indicates that while LPS drives PAR-2 expression in a dose-dependent manner, the absence of effect on GAPDH confirms that the impact is specific and not a generalized response to cellular stress.

### Evaluating the impact of PAR-2 knockdown on TNF α expression

Our investigation further delved into the consequential alterations in TNF α expression subsequent to the siRNA-mediated knockdown of PAR-2. This pivotal segment of our study was designed to test the hypothesis that suppression of PAR-2 expression would correspondingly decrease the expression of TNF α. This hypothesis stemmed from the insightful findings of Cenac et al's study, which illuminated the role of PAR-2 in triggering inflammation, and highlighted how its activation could potentiate the secretion of inflammatory mediators, notably TNF-α [70].

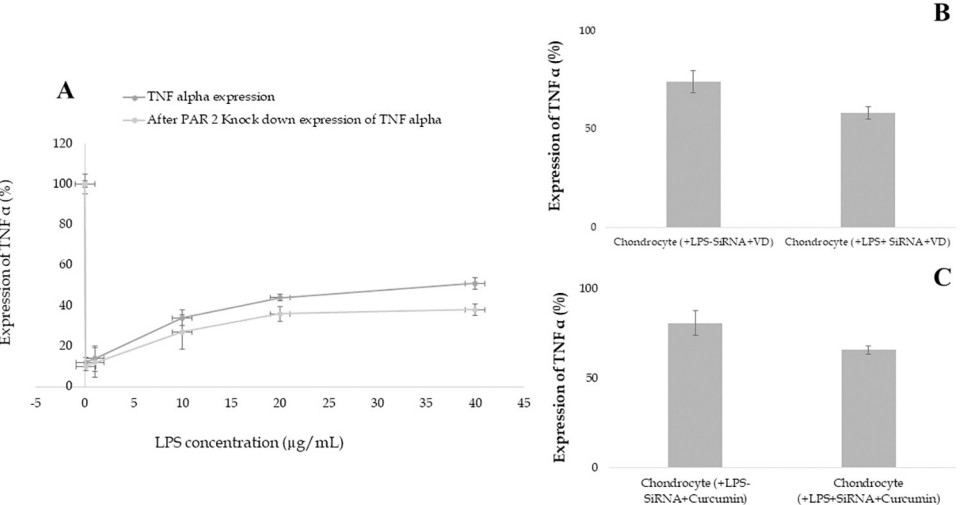

**Fig 12. Influence of siRNA-mediated PAR-2 silencing on TNF-α secretion and anti-inflammatory effects of Vitamin D and curcumin in chondrocytes.** (A) Relative secretion of TNF-α in siRNA-transfected and non-transfected chondrocytes exposed to escalating concentrations of LPS, displaying the relative decrease in TNF-α secretion upon PAR-2 knockdown and the dose-dependent increase in TNF-α levels with LPS treatment in both scenarios. (B-C) Efficacy of Vitamin D and curcumin in attenuating TNF-α expression in chondrocytes with and without PAR-2 knockdown, illustrating a reduced potency of these compounds when PAR-2 expression is silenced. Each data point represents the mean ± SD.

In our experimental setup, we compared the TNF α expression levels across varying doses of LPS in both the presence and absence of siRNA transfection, thereby providing a comprehensive perspective of TNF α's response to LPS and siRNA intervention. As depicted in Fig 12A, cells with siRNA-induced PAR-2 knockdown demonstrated a significant relative decrease in TNF α expression as compared to their non-transfected counterparts. This compelling evidence bolstered our hypothesis, indicating a crucial link between PAR-2 expression and TNF α expression. Furthermore, we observed an increase in TNF α secretion corresponding to rising doses of LPS in both experimental conditions (siRNA transfected and non-transfected). Intriguingly, Fig 12A illustrates a lesser surge in TNF α expression with increasing LPS doses in cells where PAR-2 was silenced by siRNA transfection, relative to cells where PAR-2 expression was intact. These observations together suggest that the upregulation of PAR-2 significantly impacts TNF α expression, thereby underscoring PAR-2's potential role as a regulatory switch in the inflammatory response.

Next, to better understand the interplay between PAR-2 expression and the anti-inflammatory properties of Vitamin D and curcumin, we further examined the extent to which these bioactive compounds could suppress TNF α expression following PAR-2 knockdown. As depicted in Fig 12B and 12C, our data highlighted that both Vitamin D and curcumin mitigated TNF α expression following PAR-2 knockdown. However, intriguingly, the inhibitory potency of Vitamin D and curcumin was subdued when PAR-2 expression was silenced through siRNA transfection. This observation may suggest a critical interaction between both Vitamin D and PAR-2, and curcumin and PAR-2, which likely contributes to the former's anti-inflammatory potency. It appears that in the absence or reduced presence of PAR-2, Vitamin D's and curcumin's efficacy as an inflammation suppressor might diminish, thereby underscoring PAR-2 as a possible therapeutic target for boosting both Vitamin D's and curcumin's anti-inflammatory action.

## Discussion

Inflammation is a hallmark of OA. Various strategies are availed to attenuate inflammation associated with OA. NSAIDs such as ibuprofen, naproxen, and diclofenac work by blocking the production of prostaglandins by the inhibition of cyclooxygenase (COX) enzyme [71] and have been shown to assuage inflammation in OA [72]. However, NSAIDs mediate several deleterious side effects and hence the prolonged management of OA with NSAIDs can be a veritable challenge. As NSAIDs mediate effect by blocking COX, they reduce the production of prostaglandins, which can lead to a decrease in blood flow and mucus production in the stomach, enflaming the lining of the stomach and increasing the risk of developing ulcers [73]. As NSAIDs block the production of prostaglandins, they can lead to decrease platelet aggregation and vasoconstriction, in turn increasing the risk of bleeding, which may be especially detrimental for OA subjects with bleeding disorders [74]. Prolonged use of NSAIDs can also lead to renal insufficiency especially in people with pre-existing kidney disease. In fact, this has been demonstrated by Nash et al. [75] in a large cohort study. The study investigated the association between NSAIDs and the risk of acute kidney injury (AKI) and chronic kidney disease (CKD) in a large population-based cohort. Results from the study indicated that use of NSAIDs was associated with an increased risk of AKI and CKD, especially at high doses and with long-term use [75]. Moreover, NSAIDs can cause the body to retain fluid and increase blood pressure. Johnson et al conducted a meta-analysis of 32 randomized controlled trials that investigated the effect of NSAIDs on blood pressure. The results showed that NSAIDs can cause a clinically significant increase in blood pressure, particularly in patients with pre-existing hypertension or renal disease [76]. Studies have also suggested that NSAIDs, especially at high doses, can increase the risk of heart attack and stroke, especially in people with pre-existing heart disease or other risk factors. A meta-analysis of individual participant data from randomized controlled trials of NSAIDs, including both traditional NSAIDs and selective COX-2 inhibitors (coxibs) showed that NSAID use was associated with an increased risk of major vascular events, such as heart attack and stroke, especially at high doses and with long-term use. The risk was higher in patients with pre-existing cardiovascular disease or other risk factors [77].

The other group of drugs used to manage inflammation in OA are cyclooxygenase-2 (COX-2) inhibitors such as celecoxib [10]. However prolonged use of COX-2 inhibitors is also associated with gastrointestinal problems particularly in older adults and those with a history of ulcers. In fact, in a network meta-analysis that examined the cardiovascular and gastrointestinal safety of various NSAIDs, including COX-2 inhibitors, in patients with OA or rheumatoid arthritis, found that COX-2 inhibitors were associated with an increased risk of gastrointestinal adverse events compared to placebo or non-selective NSAIDs. The risk was particularly high in older adults and those with a history of gastrointestinal ulcers [78]. Like NSAIDs (*discussed above*), long-term use of COX-2 inhibitors may increase the risk of adverse cardiovascular events. The randomized, double-blind, controlled trial VIGOR (Vioxx Gastrointestinal Outcomes Research) compared the efficacy and safety of rofecoxib (Vioxx), a COX-2 inhibitor, with naproxen, a NSAID, in patients with rheumatoid arthritis. The trial found that patients taking rofecoxib had a significantly higher risk of experiencing serious cardiovascular events, such as myocardial infarction, compared to those taking naproxen [78]. This led to the withdrawal of rofecoxib from the market in 2004 [79]. Also, like NSAIDs, COX-2 inhibitors mediate adverse renal events. In a 6-week, randomized, double-blind study involving patients with OA who were treated with fixed antihypertensive regimens and were over 65 years of age, the effects of celecoxib 200 mg/day and rofecoxib 25 mg/day on blood pressure (BP) and oedema were evaluated. Of the 1,092 patients who received study medication (celecoxib, n = 549; rofecoxib, n = 543), a significantly higher proportion of patients in the rofecoxib

group compared to the celecoxib group experienced increased systolic BP (change >20 mm Hg plus absolute value > or = 140 mm Hg) at any time point (14.9% vs 6.9%, p <0.01). The greatest increase in systolic BP in patients receiving angiotensin-converting enzyme inhibitors or beta blockers was observed with rofecoxib, while those on calcium channel antagonists or diuretic monotherapy receiving either celecoxib or rofecoxib did not show significant increases in BP. Moreover, rofecoxib was associated with a greater percentage of patients experiencing clinically significant new-onset or worsening oedema associated with weight gain (7.7%) compared to the celecoxib group (4.7%) (p <0.05) [80]. These findings raised concerns about the potential adverse effects of COX-2 inhibitors on renal function, which were further confirmed in subsequent studies [81].

In addition, to NSAIDs and COX-2 inhibitors corticosteroids such as prednisone are used for reducing inflammation associated with OA. A meta-analysis by Arroll et al. evaluated the efficacy of corticosteroid injections for the treatment of OA of the knee. The study found that corticosteroid injections were effective in reducing pain and improving physical function in patients with knee OA. However, the beneficial effects of corticosteroid injections were generally short-lived, and long-term use of corticosteroids was associated with adverse effects, such as accelerated joint degeneration and increased risk of joint infection [82]. This observation has also been confirmed in an article by Lane et al, which provides an evidence-based approach to the management of OA in the primary-care setting. The study notes that while corticosteroid injections can provide short-term pain relief, long-term use of corticosteroids can accelerate joint degeneration and increase the risk of joint infection. Based on the results, the authors recommend limiting the use of corticosteroid injections to patients with severe pain or limited mobility who have failed to respond to conservative treatments, and caution against their overuse or routine use in the management of OA [83]. Furthermore, the long-term use of corticosteroids is associated with mood change [83], weight gain [84] and increased risk of infection.

Because of the shortcomings associated with the above therapeutic intervention disease-modifying antirheumatic drugs (DMARDs)such as methotrexate and sulfasalazine, typically used to treat rheumatoid arthritis have also been assessed for treating OA and have been found effective. A consensus statement from the OA Research Society International (OARSI) regarding the management of hip and knee OA recommends the use of various treatment modalities, including DMARDs, based on an extensive review of the available evidence [85]. However, DMARDs can have significant side effects, including liver toxicity, kidney damage, and increased risk of infections. Therefore, close monitoring is necessary to ensure that patients are tolerating the medications well and to catch any potential problems early [86].

Intra-articular injection of hyaluronic acid (HA) has been shown to improve the symptoms of OA, such as pain, stiffness, and swelling. Studies have suggested that HA may have anti-inflammatory properties. Kang et al., investigated the anti-inflammatory effects of HA on chondrocyte apoptosis in a rat model of OA, where they demonstrated that HA treatment reduced chondrocyte apoptosis in the joint, suggesting that it may have anti-inflammatory effects in the context of OA [87]. Zhang et al. in a systematic review and meta-analysis analysed the results of 15 randomized controlled trials that investigated the effects of intra-articular injection of HA on inflammatory markers in patients with knee OA. The study found that HA treatment was associated with significant reductions in levels of several inflammatory markers, including IL 6, TNF α, and C-reactive protein (CRP), suggesting that it may have anti-inflammatory effects in the joint [83]. Similarly, it was investigated the effect of HA on the expression of inflammatory markers in synoviocytes obtained from patients with OA. The study found that HA treatment significantly reduced the expression of IL-1β, TNF α, and COX-2 in synoviocytes, indicating that it may have anti-inflammatory effects in the joint. Although HA is generally considered safe and have few side effects, but intra-articular injection of HA is

generally expensive and may not be covered by insurance offered by health care providers. A systematic review analyzed the results of 20 randomized controlled trials to assess the cost-effectiveness of intra-articular injections of HA for the treatment of knee OA. The study found that while hyaluronic acid injections were effective in reducing pain and improving physical function in patients with knee OA, they were generally more expensive than other treatment options, such as NSAIDs and exercise therapy [84, 85].

Therefore, nutraceuticals may present as a better alternative to NSAIDs, COX-2 inhibitors, DMARDs, and HA injections for managing inflammation in OA due to their safety, effectiveness, and affordability. In the present study, we investigated the anti-inflammatory effects of Vitamin D (1,25-dihydroxyVitamin D3 (1,25(OH)2D3)) and curcumin ((1,6-heptadiene-3,5-dione-1,7-bis(4-hydroxy-3-methoxyphenyl)-(1E,6E)) in an OA proinflammatory chondrocyte model. Both Vitamin D and curcumin exhibited potent anti-inflammatory properties especially by attenuating PAR– 2 mediated inflammation. To our knowledge, this is the first study to assess the effect of Vitamin D and curcumin on PAR2-mediated inflammation and downstream cytokines in a proinflammatory chondrocyte model mimicking OA.

Firstly, we assessed the effect of Vitamin D and curcumin on the expression of PAR-2 in the proinflammatory chondrocyte model. Both bioceuticals effectively decreased the expression of PAR-2, although the expression of the housekeeping gene used as a control was not affected (Fig 5). PAR-2 is part of the family of transmembrane, G-protein-coupled receptors and has been implicated in inducing inflammation [86], where it has been shown to induce expression of matrix metalloproteinases (MMPs) in chondrocytes leading to cartilage degradation. Proteinases are a type of enzyme that catalyze the breakdown of intact proteins by breaking peptide bonds. They play a crucial role in various scenarios, both in good health and in disease, including extracellular matrix (ECM) remodeling during development or pathological destruction. Within the human body, nearly two-thirds of the degradome's proteinases [88] operate within the extracellular space, making them ideal contenders for processes that involve ECM remodeling and/or degradation. In normal physiological conditions, the soluble collagenase subset of MMPs (MMP-1, MMP-8, and MMP-13) can break down type II collagen, which is the main component of cartilage and has a triple-helical structure. This process involves the highly specific proteolysis of a single peptide bond within the collagen molecule [89, 90]. After cleavage by these enzymes, the structure of the collagen molecule changes, becoming more susceptible to breakdown by other, less specific proteinases. This results in the disassembly of the collagen network, which is necessary during development as cartilage is converted into bone. However, this breakdown also occurs during degenerative conditions such as OA leading to the destruction of cartilage. Activation of PAR-2 induces the destruction of cartilage in a MMP dependent manner [91], whereas PAR-2 insufficiency offers protection in experimental arthritis models [92–95]. Also, PAR-2 is overexpressed in OA chondrocytes compared to normal [96], where it contributes to osteophyte formation [97]. Therefore, based on this observation, it can be stated that both Vitamin D and curcumin demonstrate the potential to effectively manage inflammation in OA by reducing PAR-2 expression (Fig 5). Consequently, incorporating these natural compounds into treatment plans for OA may be beneficial in managing inflammation and improving overall joint health. The observation that Vitamin D and curcumin are capable of downregulating the expression of MMPs—specifically MMP-1, MMP-8, and MMP-13—via reduction of PAR-2 expression (as shown in Fig 13), warrants further comprehensive investigation.

Upon activation of PAR-2, signaling through ERK1/2 and NF-κB pathways leads to the production of pro-inflammatory factors and cytokines including IL-1, TNF α, IL 6, IL 8 [98] (Fig 13). Of these pro-inflammatory cytokines TNF α has been shown to further upregulate the expression of PAR-2 in chondrocytes [98], thus initiating a vicious amplification cycle

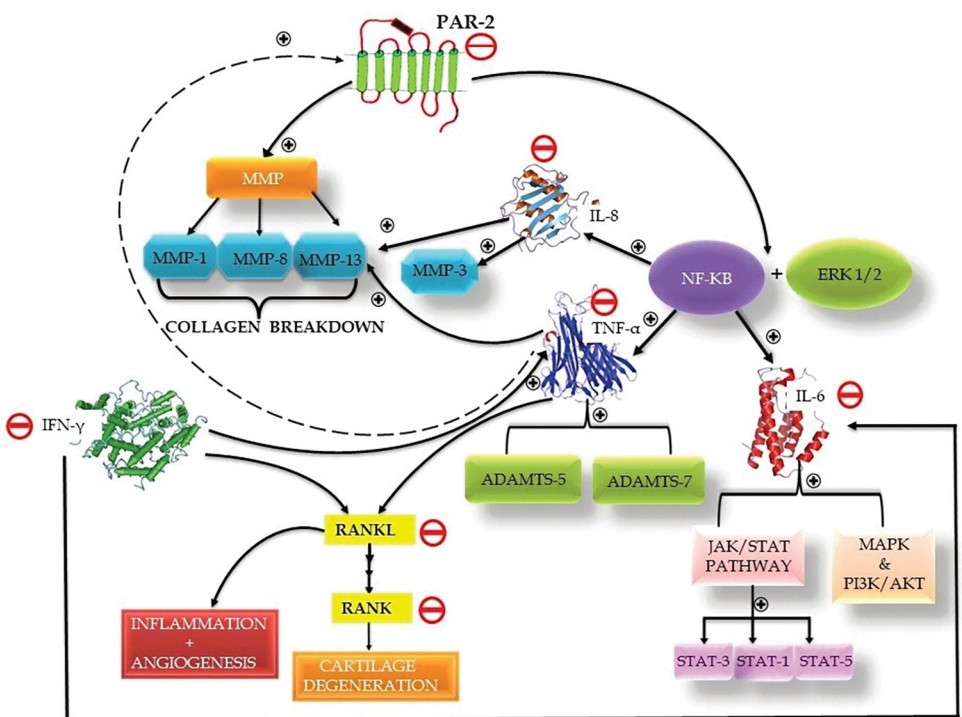

**Fig 13. Summary of the effects of Vitamin D and curcumin on PAR-2 mediated inflammation.** This diagram provides a comprehensive representation of our findings related to the mitigating effects of Vitamin D and curcumin on PAR-2 mediated inflammation in our study. Both bioceuticals reduce the expression of PAR-2 and the associated downstream proinflammatory cytokines TNF-α, Interleukin-6 (IL 6), and Interleukin-8 (IL 8), all integral to the PAR-2 mediated inflammatory pathway. We additionally observed that Vitamin D and curcumin suppress the expression of Receptor Activator of Nuclear Factor kappa-B Ligand (RANKL) and Receptor Activator of Nuclear Factor kappa-B (RANK), which are influenced by PAR-2 induced TNF α stimulation. Moreover, these compounds lessen the expression of Interferon-gamma (IFNγ), known to enhance RANKL levels. It's important to note that the anti-inflammatory action demonstrated by both Vitamin D and curcumin would naturally attenuate the pathways activated by the cytokines, the expression of which is reduced by these bioceuticals. However, as indicated in our discussion, certain aspects and potential downstream effects warrant further investigation to fully elucidate the intricate network of these pathways and their interplay with the examined bioceuticals.

(Fig 13). Therefore, in the next stage of our experiments we appraised the effect of Vitamin D and curcumin on TNF α expression. Both the bioceuticals, dose dependently inhibited the expression of TNF α (Fig 6A and 6B). TNF α, a crucial cytokine [99] involved in inflammation, is known to induce the production of various inflammatory biomarkers such as MMP-13, ADAMTS-5, and ADAMTS-7 [100, 101], in addition to activating the NF-κB signaling pathway that further exacerbates its proinflammatory effects [102]. Studies have also suggested that TNF α is implicated in cartilage degeneration associated with the development of OA [103]. It has been demonstrated in other studies that the inhibition of TNF α expression/activity can attenuate inflammation associated with OA. Zhao et al. demonstrated that exogenous administration of anti-inflammatory peptide cortistatin attenuates inflammation in primary chondrocyte culture and delays the progression of OA in spontaneous and surgically induced OA models [104]. Also, the PI3K/Akt signaling pathway is activated following TNF α exposure [105]. Han et al. evaluated the effectiveness of a combined treatment of curcumin and probucol in protecting chondrocytes from inflammatory cytokine stress in vitro and in vivo. The results showed that the combination of the two chemicals was more effective in protecting chondrocytes from stress injury induced by inflammatory cytokines than a single intervention. The combination treatment mediated its beneficial effects by blocking PI3K/Akt signaling

[106]. Since both Vitamin D and curcumin reduce PAR-2 mediated TNF α expression (shown in Fig 6A and 6B), it suggests that all subsequent pro-inflammatory processes triggered by TNF α will also be suppressed by both bioceuticals, resulting in a long-term beneficial effect on OA. However, it should be noted in the landscape of future investigations, particularly within our proinflammatory chondrocyte model, several pivotal aspects of the inflammatory signaling processes and markers merit exploration. This includes the dissection of the ERK1/2 and NF-κB signaling pathways induced by PAR-2 activation, and the resultant production of pro-inflammatory factors and cytokines. An understanding of the self-perpetuating nature of the inflammatory response, marked by the upregulation of PAR-2 expression by TNF α in chondrocytes, is also of interest. It is also critical to delineate the impact of TNF α on the induction of inflammatory biomarkers such as MMP-13, ADAMTS-5, and ADAMTS-7, and on the activation of the NF-κB signaling pathway. The potential of therapeutic strategies to inhibit TNF α expression or activity to mitigate OA-associated inflammation forms another critical facet of future exploration. Lastly, the implications of the PI3K/Akt signaling pathway upon TNF α exposure and the potential advantages of combination therapies, such as curcumin and probucol, in blocking this signaling pathway, could shed light on novel treatment strategies. These investigations will ultimately expand our understanding of the inhibitory effects of Vitamin D and curcumin on TNF α -related inflammation and their long-term benefits in managing OA, thereby offering promising new therapeutic avenues. It is to be noted that, despite extensive research, targeting TNF α and IL-1β has not yet resulted in any clinical applications for the treatment of hand and knee OA [107–110]. On the other hand, IL 6 targeting therapies have been approved and demonstrated efficacy in treating several conditions, including RA, juvenile idiopathic arthritis, Castleman's disease, and giant cell arteritis [87]. Also, in OA although IL 6 plays a significant role in joint pathology [111, 112] (including the fact that PAR-2 mediated production of IL 6) [97], it has not been a primary target of interest in research, which has instead focused predominantly on IL-1β and TNF α. In line, in this study we assessed the effect on Vitamin D and curcumin on IL 6 expression in the proinflammatory chondrocyte model. As shown in Fig 7A and 7B, both bioceuticals dose dependently inhibited IL 6 expression. *Is this outcome beneficial when it comes to the management of OA?* The IL 6 signaling pathway involves binding of IL 6 to the IL 6 receptor alpha subunit (IL 6R), followed by complex formation with a homodimer of glycoprotein 130 (gp130) leading to activation of the Janus kinases/ signal transducers and activators of transcription (JAK/STAT) pathway (Fig 13), which in turn leads to recruitment and activation of STAT1, STAT3, and to a lesser extent STAT5 [113, 114] (Fig 13). IL 6 also activates non-canonical signaling via the mitogen-activated protein kinase (MAPK) cascade and PI3K-protein kinase B (PkB)/Akt (Fig 13). Negative feedback regulators, such as the suppressor of cytokine signalling (SOCS) protein family and protein inhibitors of activated STATs (PIAS), tightly control IL 6-induced JAK/STAT signaling. SOCS3 is a specific inhibitor of IL 6 signaling and directly inhibits JAK-kinase activity, while PIAS negative inhibitors inhibit DNA-binding activity by binding to activated STAT-dimers [115–118]. Also, IL 6 has two modes of receptor activation: classic signaling through membrane-anchored IL 6R [119] and trans-signaling through soluble IL 6R (sIL 6R) [120]. IL-6 levels are significantly increased in synovial fluid and serum of OA subjects [111]. Also, several experimental OA models have shown that IL 6 induces catabolic mediators in OA, leading to cartilage destruction [121]. Inhibition of IL 6 using neutralizing antibodies or siRNA resulted in decreased cartilage lesions and subchondral bone sclerosis in OA models [122], while treatment with anti-IL 6-receptor neutralizing antibodies resulted in amelioration of cartilage pathology, synovial inflammation, and osteophyte development [123]. Tocilizumab, a drug that targets the human IL 6R [124], was also effective in preserving cartilage in a mouse model of ischemic osteonecrosis and increasing bone volume [125]. Hence, the fact that both Vitamin D and curcumin

reduce IL 6 expression in our chondrocyte model of proinflammation alludes to the fact that these bioceuticals will mediate anti-inflammation in OA leading to beneficial outcomes. However, how these bioceuticals will affect the expression of sIL 6R, which controls activation of IL 6 trans-signaling, warrants further research. Also, there are preclinical indication that the inhibition of JAK/STAT signaling is effective in attenuating inflammation in OA. An example is when tofacitinib prevented cytokine-induced proteoglycan loss and reinstated collagen type II synthesis in bovine cartilage explants [126]. Furthermore, animal research has shown that inhibiting JAK/STAT has protective effects in experimental OA. How, Vitamin D and curcumin will affect JAK/STAT signaling also needs to be investigated, which can form the basis of further future studies. That being said, JAK/STAT targeting also hinders the function of several cytokines, such as IL-10, IL-4, and IGF-1, which play a positive role in joint biology and OA development [127–129]. Because OA is an extremely diverse condition with significant differences in inflammation severity and cytokine profile, predicting the outcome of JAK/STAT inhibition in OA patients will be challenging. Hence, opting for a more straightforward method of targeting a single cytokine, such as IL 6, could be a more secure approach, which is why we restricted ourselves only to IL 6 in the present study. Future investigations utilizing our proinflammatory chondrocyte model should place significant emphasis on understanding how bioceuticals, like Vitamin D and curcumin, affect the complex signaling processes mediated by IL-6. Given the intricate IL-6 signaling pathway, which encompasses JAK/STAT activation, MAPK cascade, PI3K-PkB/Akt, and mechanisms of negative feedback regulation, it is crucial to unravel how the reduction of IL-6 expression by these bioceuticals impacts these cascades. Furthermore, the different modes of IL-6 receptor activation, particularly the role of soluble IL-6 receptor in IL-6 trans-signaling, warrant comprehensive exploration. Another focal point should be the evaluation of these bioceuticals on the JAK/STAT signaling pathway, keeping in mind its broad role in the function of multiple cytokines and its potential diverse effects given the heterogeneity of OA. While inhibition of this pathway has shown promising effects in preclinical models, understanding its specific implications when IL-6 is targeted by Vitamin D and curcumin could provide insightful avenues for further research.

As indicated earlier, activation of PAR-2 induces IL 8 expression [130] (Fig 13). IL 8, also known as CXCL-8, is an inflammatory chemokine that is produced by chondrocytes in individuals with OA. IL 8 is believed to play a critical role in the development and progression of OA by promoting the release of MMP-13 and contributing to inflammatory changes in the synovium. Previous studies have demonstrated that IL 8 can induce hypertrophy and differentiation of chondrocytes in OA using cartilage culture [131]. In patients with OA, synovial fluid levels of IL 8 are elevated compared to younger individuals with anterior cruciate ligament injury, and elevated levels of IL 8 have been associated with MMP-3 activation in plasma [132]. A cross-sectional study by García-Manrique et al. investigated the association between IL 8 and clinical severity, as well as local and systemic adipokines and cytokines, in 115 women with symptomatic knee OA. The study found that synovial fluid IL 8 was significantly associated with clinical severity scales and was also associated with TNF, IL6, osteopontin, visfatin, and resistin. However, no associations were detected in serum, suggesting that local inflammation is more relevant to clinical severity in knee OA with joint effusion [133]. In line, in the next stage of our study we assessed the effect of the bioceuticals on the expression of IL 8. Both Vitamin D and curcumin dose dependently attenuated the expression of IL 8 (Fig 8A and 8B). The diverse effects of IL8 may provide an explanation for its involvement in OA. These effects include attracting neutrophils, stimulating the activation and migration of leukocytes to the joint, and directly affecting the hypertrophy and differentiation of chondrocytes or increasing the release of matrix metalloproteinases [134]. IL8 may also be linked to chronic inflammation-related angiogenic changes [135]. Our study results have shown that Vitamin D and

curcumin can decrease the expression of IL 8 in a pro-inflammatory chondrocyte model. Since previous research suggests that IL 8 is primarily involved in local inflammation associated with OA, rather than systemic inflammation, the findings point towards the potential of using these bioceuticals to develop effective strategies for managing local inflammation in OA patients. In summary, the ability of Vitamin D and curcumin to target IL 8 in the pro-inflammatory chondrocyte model suggests that they could be effective in ameliorating the local inflammation commonly associated with OA. This highlights the potential of using these bioceuticals as a key direction for developing new therapeutic strategies for OA patients.

The signaling system composed of RANK(receptor activator of nuclear factor kappa-B), RANKL (receptor activator of nuclear factor kappa-B ligand), and OPG (osteoprotegerin) is known for its various roles in bone remodeling, modelling, and osteoclast maturation. Although its main components are primarily involved in bone homeostasis, the RANKL/ RANK/OPG pathway has been found to have other effects as well [136]. RANKL is a protein that exists as a homotrimer and is generated by osteoblasts and activated T cells [137]. The secretion of RANKL occurs either through proteolytic cleavage or alternative splicing from the membrane form [138]. Proteolytic cleavage of RANKL is carried out by enzymes like MMP3 or 7, or ADAM [139]. RANKL, which is secreted by preosteoblasts, osteoblasts, osteocytes, and periosteal cells, activates RANK, which is expressed by osteoclasts and their precursor cells [140]. RANKL plays a crucial role in preosteoclast differentiation [141], adherence of osteoclasts to bone tissue [142], activation [143], and maintenance [144]. However, the specific mechanism by which RANKL affects the fusion of preosteoclasts to form multinucleated cells is still unclear. RANK belongs to the TNF family of homotrimeric transmembrane receptors. It is primarily expressed in OPCs, dendritic cells, and mature osteoclasts [145], but unlike other TNF family receptors, it lacks intrinsic protein kinase activating activity. While all TRAFs 2, 5, and 6 bind to RANK, only TRAF 6 is essential for maintaining bone health [146–148]. In summary, RANK and RANKL are signaling molecules that play important roles in bone metabolism and are involved in the pathogenesis of OA [149]. The binding of RANKL to RANK promotes osteoclast differentiation, activation, and survival, leading to bone resorption [150, 151] (Fig 13). In OA, increased expression of RANKL in synovial fibroblasts and chondrocytes leads to excessive bone resorption, resulting in subchondral bone sclerosis and cartilage degeneration [152, 153]. Furthermore, studies have shown that RANKL is involved in the regulation of synovial inflammation and angiogenesis in OA [154]. Studies have also suggested that RANKL is involved in the crosstalk between bone and cartilage in OA [155]. Increased subchondral bone turnover mediated by RANKL leads to the release of factors that promote cartilage degradation, such as matrix metalloproteinases and cytokines [156]. This interaction between bone and cartilage contributes to the pathogenesis of OA. Targeting the RANK/ RANKL pathway has been proposed as a potential therapeutic strategy for OA. Inhibition of RANKL using monoclonal antibodies or small molecule inhibitors has been shown to reduce subchondral bone resorption and cartilage degradation in animal models of OA [157]. Further to the above, TNF α induces the expression of RANKL in chondrocytes [158], which can activate RANK on the surface of the osteoclasts and promote bone resorption in the joints (Fig 13). Since, both Vitamin D and curcumin attenuate expression of TNF α (shown in Fig 6A and 6B), we also investigated the effect of these bioceuticals on the expression of RANK and RANKL in our proinflammatory chondrocyte model using flowcytometry. Both Vitamin D and curcumin decreased the expression of RANK and RANKL in our proinflammatory model (shown in Fig 10), suggesting a new strategy for the treatment/management of OA. Building on our findings, potential future investigations could explore the precise role of RANKL in the fusion of preosteoclasts, and the mechanisms by which Vitamin D and curcumin impact the RANK/RANKL pathway. Moreover, evaluating the therapeutic effectiveness of RANKL

inhibition by these bioceuticals in animal models of OA could provide valuable insights into the development of targeted therapies for OA.

Another cytokine that has been implicated in OA is *IFN γ*. The levels of *IFN γ* are elevated in the synovial fluids of patients who have reached the end stage of OA [67]. Additionally, treating chondrocytes with *IFN γ* propagates inflammatory and degenerative events due to an increase in the transcription of important inflammatory mediators such as TNF α and IL 6 via protein kinase R (PKR) activation [159]. Furthermore, interferon-gamma can promote the expression of RANKL in chondrocytes [160]. Therefore, *IFN γ* presents itself as an important therapeutic target for the treatment of OA. In line, with this premise we assessed the effect of Vitamin D and curcumin on the expression of *IFN γ* in our proinflammatory chondrocyte model. Both bioceuticals exhibited a dose dependent reduction in the expression of interferon-gamma (Fig 9A and 9B). Considering the increased levels of *IFN γ* and its role in promoting inflammatory and degenerative events in OA, including the promotion of RANKL expression, an intriguing future study might involve examining the interplay between IFN γ, RANKL/ RANK pathway, and the effect of bioceuticals, Vitamin D and curcumin. Utilizing our established proinflammatory chondrocyte model, the study could assess how Vitamin D and curcumin modulate the expression of IFN γ and RANKL/RANK, and in turn, influence the inflammatory and bone resorption processes in OA. This could provide a more comprehensive understanding of the multi-faceted interactions in OA pathogenesis and potentially identify more targeted treatment strategies.

As our investigations indicated that down regulation of PAR– 2 is a crucial step in the mediation of anti-inflammation by both Vitamin D and curcumin (Fig 5), we assessed the anti-inflammation potency of the bioceuticals by regulating the expression of PAR– 2. We initiated our study by silencing PAR-2 expression in chondrocytes under LPS stimulation, using siRNA technology (Fig 10). This provided us with an opportunity to assess the direct role of PAR-2 in mediating the inflammatory response, particularly the expression of TNF α (Fig 11). We observed a substantial reduction in TNF α levels following the PAR-2 knockdown, underscoring the potential of PAR-2 as a therapeutic target in OA (Fig 12). Further, we evaluated the modulatory effect of Vitamin D and curcumin on the expression TNF α post-PAR-2 knockdown through siRNA transfection (Fig 12B and 12C). Although, both bioceuticals decreased TNF α expression, there was diminished efficacy of these bioactives in the presence of PAR-2 knockdown (Fig 12B and 12C). However, we observed that the TNF α expression was not completely abrogated, despite successful PAR-2 silencing and the application of Vitamin D and curcumin (Fig 12). The incomplete abrogation of TNF α expression even when PAR-2 is silenced suggests that there are other molecular pathways at play that can induce TNF α independently of PAR-2. The complex nature of inflammatory responses involves multiple overlapping and interconnected pathways [71]. TNF α is a pivotal cytokine in inflammatory responses and its production can be stimulated by a variety of molecular signals [161]. These include other cytokines (like IL-1 and IL-6), growth factors, viral or bacterial components, and even physical stress. Similarly, activation of the NLRP3 inflammasome, a multi-protein complex that senses cellular stress signals, can also lead to the production of TNF α [162]. Furthermore, TNF α expression is regulated by several transcription factors such as nuclear factor-kappa B (NF-κB) and activator protein-1 (AP-1), which can be activated by multiple signaling pathways independent of PAR-2. Therefore, even when PAR-2 is silenced, these other pathways can still lead to the production of TNF α. It's also worth noting that while siRNA silencing can effectively reduce the expression of a targeted gene, it rarely completely eliminates its expression [163], and residual PAR-2 may still be able to contribute to TNF α production. Thus, while PAR-2 is a significant contributor to the inflammatory response in OA, it is just one component of a complex network of signaling pathways and molecular interactions that

regulate the production of pro-inflammatory cytokines like TNF α. These multiple, overlapping pathways provide redundancy in the system, ensuring that inflammatory responses can still be activated even if one pathway is disrupted, which is likely why we see persistent TNF α expression despite PAR-2 silencing.

In conclusion, our study demonstrates that both Vitamin D and curcumin can attenuate the pro-inflammatory response in chondrocytes by inhibiting PAR-2 signaling, reducing the expression of TNF α, IL 6, and IL 8, as well as the RANKL/RANK system (Fig 13). Moreover, these bioceuticals also reduce *IFN γ* expression, which amplifies the inflammatory events in OA (Fig 13). These findings suggest that Vitamin D and curcumin have potential therapeutic benefits in the management of OA.

As indicated above, clinical trials investigating the efficacy of Vitamin D in OA have yielded mixed results (Table 1) although, our study alludes to the fact that Vitamin D should have therapeutic benefits in the management of OA. Overall, the mixed results obtained with Vitamin D in OA trials can be attributed to the heterogeneity of study populations, variability in dosing and duration, lack of consistency in outcome measures, and confounding factors. More well-designed clinical trials are needed to fully understand the potential role of Vitamin D in treating OA.

As far as our knowledge goes, this is the first study to showcase the potential therapeutic benefits of curcumin in managing OA. However, there are several challenges associated with curcumin that may limit its therapeutic potential in OA. Firstly, curcumin is poorly soluble in water, which limits its bioavailability and makes it difficult to deliver in therapeutic doses. Although, various approaches to enhance the solubility of curcumin, including the use of nanoparticles, liposomes, and cyclodextrins have been availed, the use of these strategies for the treatment/management of OA needs to be assessed [33]. Secondly, curcumin undergoes extensive metabolism in the body, which further limits its bioavailability. Some studies have shown that combining curcumin with piperine, a compound found in black pepper, can increase its bioavailability by inhibiting metabolism. This needs to be assessed in suitable preclinical OA models [33].

While our study provides valuable insights into the potential therapeutic benefits of Vitamin D and curcumin in the management of OA, there are several areas where further research needs to be conducted. Firstly, the study was conducted on a proinflammatory chondrocyte model, and it would be advantageous to validate these findings in a preclinical OA model to understand the effects of these bioceuticals on the disease progression *in vivo*. Secondly, the study investigated the effect of Vitamin D and curcumin on the RANKL/RANK system, which plays an imperative role in the development of OA. However, further studies are needed to investigate the underlying mechanisms by which these bioceuticals exert their effects on this system. Thirdly, the study evaluated the effect of Vitamin D and curcumin on *IFN γ* expression, which is known to amplify the inflammatory events in OA. However, it would be worthwhile to explore the effect of these bioceuticals on other inflammatory mediators, chemokines, and matrix metalloproteinases. Fourthly, the study did not investigate the effects of Vitamin D and curcumin on cartilage degradation or joint function in OA. Further studies could investigate the effects of these bioceuticals on these important clinical outcomes employing suitable *in vitro* and *in vivo* models. Finally, the study did not investigate the optimal dose or duration of treatment with Vitamin D and curcumin, and future studies could aim to address these questions to optimize their therapeutic efficacy in OA.

## Acknowledgments

The authors acknowledge the support provided by MBRU through a grant which enabled the conduction of this study. The authors will like to acknowledge the kind help of Ms Shirin

Jannati in editing the bibliography and also in the preparation of the final version of the manuscript. Additionally, SR is grateful to be a recipient of a stipend under the Biomedical Science Post Graduate program at MBRU. This work was supported in part by the Al Jalila Foundation.

## Author Contributions

**Conceptualization:** Rajashree Patnaik, Yajnavalka Banerjee.

**Data curation:** Rajashree Patnaik, Sumbal Riaz, Bala Mohan Sivani, Shemima Faisal, Yajnavalka Banerjee.

**Formal analysis:** Rajashree Patnaik, Nerissa Naidoo, Manfredi Rizzo, Yajnavalka Banerjee.

**Funding acquisition:** Nerissa Naidoo, Yajnavalka Banerjee.

**Investigation:** Yajnavalka Banerjee.

**Methodology:** Rajashree Patnaik, Sumbal Riaz, Bala Mohan Sivani, Yajnavalka Banerjee.

**Supervision:** Nerissa Naidoo, Yajnavalka Banerjee.

**Validation:** Rajashree Patnaik, Nerissa Naidoo, Yajnavalka Banerjee.

**Writing – original draft:** Rajashree Patnaik, Yajnavalka Banerjee.

**Writing – review & editing:** Yajnavalka Banerjee.

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
