## [Decision Letter · Decision Letter 0]

10 Apr 2023

PONE-D-23-07167Evaluating the potential of vitamin D and curcumin to alleviate inflammation and mitigate the progression of osteoarthritis through their effects on human chondrocytes: a proof-of-concept investigation.PLOS ONE

Dear Dr. Banerjee,

Thank you for submitting your manuscript to PLOS ONE. After careful consideration, we feel that it has merit but does not fully meet PLOS ONE’s publication criteria as it currently stands. Therefore, we invite you to submit a revised version of the manuscript that addresses the points raised during the review process.

We look forward to receiving your revised manuscript.

Kind regards,

Andre van Wijnen

Academic Editor

PLOS ONE

Journal Requirements:

   "This study is supported by research award from Mohammed Bin Rashid University of Medicine and Health Sciences (MBRU), jointly awarded to NN and YB with the following identifier; MBRU-CM-RG2022-04.SR is a recipient of scholarship in the Biomedical Sciences Master Program at MBRU."

Reviewers' comments:

Reviewer's Responses to Questions

**Comments to the Author**

1. Is the manuscript technically sound, and do the data support the conclusions?

Reviewer #1: Partly

2. Has the statistical analysis been performed appropriately and rigorously? 

Reviewer #1: Yes

3. Have the authors made all data underlying the findings in their manuscript fully available?

Reviewer #1: Yes

4. Is the manuscript presented in an intelligible fashion and written in standard English?

Reviewer #1: No

5. Review Comments to the Author

Reviewer #1: 1. The author refers to a large number of references in the INTRODUCTION. However, for the research article, the background information related to the research should be clarified succinctly and clearly.

2. In the Results section, the authors only considered the anti-inflammatory effects of vitamin D and curcumin in the proinflammatory chondrocyte model, but did not consider the toxic effects of vitamin D and curcumin on chondrocytes, which is a serious defect.

3. The authors considered that PAR-2 expression is critical for the anti-inflammatory effects of vitamin D and curcumin on chondrocytes. Why not test this conclusion by regulating the expression of PAR-2?

4. The author's summary presented in Figure 10 is based on references, but most of the conclusions have not been verified in this paper

6. PLOS authors have the option to publish the peer review history of their article (what does this mean?). If published, this will include your full peer review and any attached files.

Reviewer #1: No

---

## [Author Response · Author response to Decision Letter 0]

6 Jul 2023

Subject: Response to Review Comments for Manuscript ID PONE-D-23-07167: "Evaluating the potential of vitamin D and curcumin to alleviate inflammation and mitigate the progression of osteoarthritis through their effects on human chondrocytes: a proof-of-concept investigation"

Dear Professor Andre van Wijnen,

We greatly appreciate your careful review and constructive feedback regarding our manuscript titled "Evaluating the potential of vitamin D and curcumin to alleviate inflammation and mitigate the progression of osteoarthritis through their effects on human chondrocytes: a proof-of-concept investigation". 

We have taken your valuable suggestions into account and have revised our manuscript accordingly. We have taken your valuable suggestions into account and made extensive revisions to our manuscript. All changes introduced in the revised manuscript are clearly indicated in the form of track changes. The details of our responses to your comments and the changes made to our manuscript are as follows:

Comment: 1

The author refers to a large number of references in the INTRODUCTION. However, for the research article, the background information related to the research should be clarified succinctly and clearly.

Response

We appreciate your suggestion to provide a more succinct and clear background. We understand the need for clarity, conciseness, and relevance in setting up the context for our research work.

In response to your comments, we have extensively revised the introduction (refer to the manuscript with track changes for details) to ensure a balance between detail and brevity. We have retained the original number of references as each of them serves a specific purpose in elucidating our context and shaping the premise of our study. Each reference provides a piece of the complex puzzle of understanding the roles and potential of drugs targeting proinflammatory cytokines, a field that is diverse and extensive. Moreover, the references provide the necessary backdrop to comprehend the therapeutic potential of vitamin D and curcumin in the context of osteoarthritis (OA), creating a comprehensive picture of the current landscape of knowledge in this area. We believe that the comprehensiveness of our citation will aid readers in exploring the topic further if they wish to, and thus, it adds value to the overall understanding of the research field.

Regarding the table reviewing the studies assessing the effect of vitamin D in OA, we believe it is of paramount importance to our research. This table synthesizes a wide array of studies and gives readers a thorough overview of the state of current research in a concise and easily digestible format. By summarizing the outcomes and methodologies of several studies, it enables readers to quickly grasp the collective findings on the impact of vitamin D on OA. This, in turn, underscores the novelty and significance of our study, which extends this knowledge by investigating the role of vitamin D in the context of PAR-2 mediated inflammation in OA.

In conclusion, while we have worked expansively and extensively on the brevity and clarity of the introduction, we believe that the breadth of references and the inclusion of the review table add significant value to our research article, offering the readers a comprehensive view of the field and our unique contribution to it.

Comment: 2

In the Results section, the authors only considered the anti-inflammatory effects of vitamin D and curcumin in the proinflammatory chondrocyte model, but did not consider the toxic effects of vitamin D and curcumin on chondrocytes, which is a serious defect.

Response

Thank you for your insightful comments. We recognize the importance of considering the toxic effects of Vitamin D and curcumin on chondrocytes and we wholeheartedly agree with you on its importance.

In our initial submission, we noted that both Vitamin D and curcumin did not affect the expression of the control glyceraldehyde 3-phosphate dehydrogenase (GAPDH) gene in our Western blot experiments. This observation suggested that these compounds were most likely non-cytotoxic at the doses used. However, we understand the need for a more direct assessment of cytotoxicity.

To strengthen our study and address your valuable comment, we have now performed further cytotoxicity evaluations using the MTT (3-(4,5-Dimethylthiazol-2-yl)-2,5-diphenyltetrazolium bromide) assay, a widely recognized method for its robustness in cytotoxicity evaluation. This assay works by measuring the metabolic activity of the cells, thereby providing a reliable measure of cell health and proliferation.

We have tested various doses for each compound: LPS (0.1 µg/mL, 1 µg/mL, 10 µg/mL, 20 µg/mL, and 40 µg/mL), Vitamin D (0.01 µM, 0.02 µM, 0.12 µM, 0.25 µM), and Curcumin (10 µM, 25 µM, 50 µM, 75 µM, 100 µM). The selected dose ranges for LPS encompasses the dose that was used to create the proinflammatory chondrocyte model. For Vitamin D, and curcumin the used concentrations encompass the doses used in our ELISA, Western Blot, and Flow cytometry studies, where both compounds showed anti-inflammatory effects.

Our findings from the MTT assay indicate no observable cytotoxicity at any of the tested concentrations, as illustrated in the newly added Figure 3 in the revised version of the manuscript. This vital observation underscores that the doses of LPS, Vitamin D, and curcumin used in our experiments do not exert cytotoxic effects and should not interfere with the cells' normal function, further reinforcing their potential as safe therapeutic agents in the study and management of OA.

To reflect these important additions, we have revised the Methods, Results and Discussion sections of our manuscript (identifiable through track changes in the track-changed manuscript). In the Discussion, we specifically address the lack of observed cytotoxicity for Vitamin D and curcumin at the tested doses and provide context for these results in the broader landscape of OA therapeutics. These changes are also highlighted in the manuscript through track changes.

We hope that these additions and modifications adequately address your concerns and further validate our study's findings.

Comment: 3

The authors considered that PAR-2 expression is critical for the anti-inflammatory effects of vitamin D and curcumin on chondrocytes. Why not test this conclusion by regulating the expression of PAR-2?

Response

Thank you for your insightful comments and the opportunity to further clarify our experimental approach. We appreciate your suggestion of testing our hypothesis regarding the critical role of PAR-2 in the anti-inflammatory effects of vitamin D and curcumin. In fact, in response to your comment, we have performed additional experiments to directly assess this hypothesis.

To delineate the role of PAR-2 expression in regulating the anti-inflammatory effects of vitamin D and curcumin, we used siRNA to knockdown the expression of PAR-2 in our chondrocyte model. As you've suggested, we evaluated the consequential changes in TNF-α expression under various conditions of LPS treatment in the absence and presence of siRNA transfection. Our findings (please refer to the revised Results section and Figures 10 and 11 in the manuscript) corroborate the hypothesis that diminished PAR-2 expression results in a relative decrease in the secretion of TNF-α, affirming the role of PAR-2 in mediating inflammatory responses in chondrocytes.

However, it's worth noting that even with the knockdown of PAR-2 expression, we did not observe a complete abrogation of TNF-α secretion. This observation suggests that although PAR-2 plays a significant role in mediating the inflammatory response, other signalling pathways or receptors might also be involved in the regulation of TNF-α secretion. Therefore, the anti-inflammatory effects of vitamin D and curcumin may also target these alternative pathways, offering a multifaceted mode of action against inflammation.

Additionally, we tested the effect of vitamin D and curcumin on PAR-2 silenced chondrocytes. The diminished efficacy of these bioactives in the presence of PAR-2 knockdown offers further evidence towards their targeted activity against PAR-2-mediated inflammation.

These additions have been incorporated into our methodology, results, and discussion, which you will find marked clearly with track changes in the revised manuscript.

We believe that these new experiments strengthen our study by directly demonstrating the role of PAR-2 in mediating the anti-inflammatory effects of vitamin D and curcumin. We hope that this addresses your concerns and further reinforces the significance of our findings.

Comment: 4

The author's summary presented in Figure 10 is based on references, but most of the conclusions have not been verified in this paper

Response

We appreciate the reviewer's feedback and the opportunity to further clarify the importance of Figure 10 (Figure 13 in the revised manuscript). This figure serves as a comprehensive summary of our findings and how they contribute to the larger landscape of OA-associated inflammation, outlining the role of key proinflammatory cytokines and signalling molecules, and the modulatory effects of bioceuticals like vitamin D and curcumin.

Firstly, Figure 10 provides an integrated view of the intricate relationships between PAR-2 and other key cytokines such as TNF-alpha, IL-6, IL-8, and IFNgamma, as well as crucial signalling molecules like RANK and RANKL, all known to play significant roles in OA pathogenesis, and investigated in the present study.

Secondly, this figure illustrates the impact of vitamin D and curcumin on these pathways. Both bioceuticals exhibited significant anti-inflammatory properties by reducing the expression of these proinflammatory mediators and hence, can be considered as promising agents for OA management.

Finally, Figure 10 serves as a roadmap for future research. While it's true that not all depicted relationships and mechanisms have been verified within the scope of our current study, each of these points represents a significant and promising area for further investigation and have been indicated in the discussion of our manuscript in the form of track changes. Understanding the multiple layers of interactions between these cytokines and signalling molecules in the context of OA inflammation is a challenging but crucial task. This figure sets the stage for future research endeavours to delve deeper into the unexplored areas of this complex network.

We believe that our study, alongside Figure 10, significantly contributes to the broader understanding of OA inflammation. Furthermore, it highlights the potential therapeutic benefits of natural bioceuticals and underlines the need for continued research into the complex interplay of inflammatory mediators in OA.

We believe the modifications made in response to your comments have significantly strengthened our manuscript. We appreciate your time and consideration and look forward to hearing your thoughts on our revised submission.

Kindly do not hesitate to approach my office if you require further information or clarification.

Looking forward to a positive response from your side at the earliest,

Regards and Best Wishes,

Yajnavalka Banerjee, PhD, PGDME

Associate Professor of Biochemistry

---

## [Editor Report · Decision Letter 1]

15 Aug 2023

Evaluating the potential of vitamin D and curcumin to alleviate inflammation and mitigate the progression of osteoarthritis through their effects on human chondrocytes: a proof-of-concept investigation.

PONE-D-23-07167R1

Dear Dr. Banerjee,

We are pleased to inform you that your manuscript has been judged scientifically suitable for publication and will be formally accepted for publication once it meets all outstanding technical requirements.

Within one week, you will receive an e-mail detailing the required amendments. When these have been addressed, you’ll receive a formal acceptance letter and your manuscript will be scheduled for publication.

Kind regards,

Andre van Wijnen, PhD

Academic Editor

PLOS ONE

Additional Editor Comments (optional):

The authors have adequately revised this paper in response to the previous critiques.

Reviewers' comments:

Prevous reviewers did not provide comments.

---

## [Editor Report · Acceptance letter]

20 Sep 2023

PONE-D-23-07167R1 

Evaluating the potential of vitamin D and curcumin to alleviate inflammation and mitigate the progression of osteoarthritis through their effects on human chondrocytes: a proof-of-concept investigation. 

Dear Dr. Banerjee:

I'm pleased to inform you that your manuscript has been deemed suitable for publication in PLOS ONE. Congratulations! Your manuscript is now with our production department. 

Kind regards, 

on behalf of

Dr. Andre van Wijnen 

Academic Editor

PLOS ONE